# Scalable Transformer for PDE Surrogate Modeling

**Zijie Li, Dule Shu, Amir Barati Farimani**
Carnegie Mellon University
Mechanical Engineering Department
{zijieli, dules}@andrew.cmu.edu & barati@cmu.edu

## Abstract

Transformer has shown state-of-the-art performance on various applications and has recently emerged as a promising tool for surrogate modeling of partial differential equations (PDEs). Despite the introduction of linear-complexity attention, applying Transformer to problems with a large number of grid points can be numerically unstable and computationally expensive. In this work, we propose Factorized Transformer (FactFormer), which is based on an axial factorized kernel integral. Concretely, we introduce a learnable projection operator that decomposes the input function into multiple sub-functions with one-dimensional domain. These sub-functions are then evaluated and used to compute the instance-based kernel with an axial factorized scheme. We showcase that the proposed model is able to simulate 2D Kolmogorov flow on a $256 \times 256$ grid and 3D smoke buoyancy on a $64 \times 64 \times 64$ grid with good accuracy and efficiency. The proposed factorized scheme can serve as a computationally efficient low-rank surrogate for the full attention scheme when dealing with multi-dimensional problems.

## 1  Introduction

Various physics processes are modeled by partial differential equations (PDEs), from the interaction between atoms in molecular systems to large-scale cosmological phenomena. Solving PDEs advances the understanding of complex physical phenomena, enabling people to make accurate predictions, and make informed decisions across a wide range of scientific and engineering disciplines. Numerical solvers provide a practical way to simulate and predict PDEs since many PDEs are often difficult to solve analytically. Most numerical solvers divide the continuous domain into a discretized grid and reduce the continuous differential equations to algebraic equations via methods like finite difference/element/volume methods or spectral method. Despite the theoretical guarantees behind them, their practical realization of specific problems can pose challenges that require careful expertise to overcome, such as a sufficient understanding of the underlying physics, or a fine-tailored mesh that resolves the necessary spatio-temporal scales. The interest in developing user-friendly and efficient PDE solvers, along with the success of deep learning models in many other areas [13, 44, 51, 103], has facilitated the emergence of neural-network-based PDE solvers, where the neural network can be used to parameterize the solution function of the target equation [96], or to approximate the solution operator[69, 76]. Compared to many numerical solvers, neural PDE solvers appear to be more tolerant with coarse discretization [104], and can be applied without explicit meshing [96]. In addition, knowing the underlying equations are not strictly necessary for neural PDE solvers, which gives them the potential to simplify and accelerate the process of physics simulation based on PDEs.

Among various neural network designs, attention-based models (Transformer) [115] have become state-of-the-art for a wide array of applications [13, 18, 27, 51], which gives rise to a recent surge of interest in applying the Transformer to PDE modeling [16, 29, 35, 41, 43, 54, 67, 73, 82, 86]. By viewing the input sequence as a function sampled on a discretization grid, attention can be interpreted as a learnable kernel integral [17, 35, 54, 60] or a learnable Galerkin projection [16], and the sequence-to-sequence Transformer [115] can be modified correspondingly to be better suited for

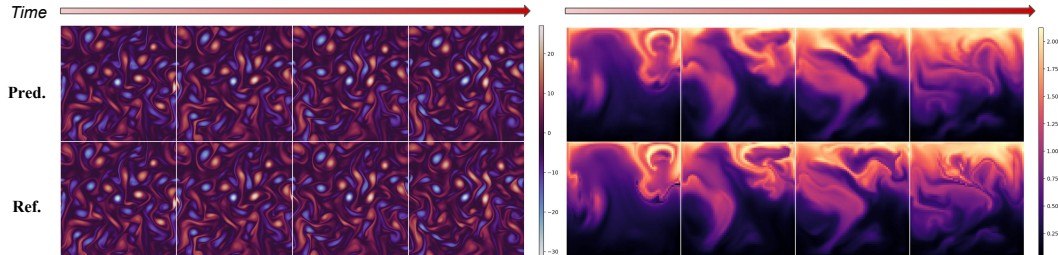

Figure 1: Model's prediction (pred.) and reference ground truth (ref.). **Left**: 2D Kolmogorov flow on $256 \times 256$ grid; **Right**: 3D smoke buoyancy on $64 \times 64 \times 64$ grid ($zOy$ cross-section is shown).

PDE modeling [16, 35, 43, 54, 67, 74, 86]. In these works, attention is typically applied to every grid point in the domain to exploit both the local and non-local structure of the system, and therefore a linear-complexity variant of attention is usually necessary. As the number of grid points grows exponentially with respect to the number of dimensions, this results in a very large attention matrix that computes the interaction between every pair of the grid points (despite this attention matrix is not evaluated explicitly in linear attention). Consequently, cascading a deep stack of attention layers introduces instability and relatively high computational cost on high-resolution grid. To alleviate these issues and improve the scalability of Transformer in PDE modeling, we propose a modified attention mechanism. Our model is inspired by the kernel integral viewpoint of softmax-free attention, with a factorized integration scheme motivated by the inherent low-rank structure of dot-product kernel matrix. More specifically, we propose a multi-dimensional factorized kernel integral with each kernel function in the integral having only single-dimensional domains. To calculate these axial kernels, we propose a learnable integral operator that is able to project the input function with high-dimensional domain into a set of sub-functions with single-dimensional domain. The computation of each axial kernel is quadratic with respect to the number of grid points along that the corresponding axis but does not grow with the number of dimensions, which alleviates the curse of dimensionality in standard attention. With the modified attention mechanism, our proposed model can scale up to multi-dimensional problems with a large number of grid points and achieve competitive performance compared to state-of-the-art models. Moreover, we show that our factorized attention mechanism can reduce the computational cost and improve stability compared to softmax-free linear attention.[1]

## 2   Related works

**Neural PDE solver**   Based on the emphases of model design, neural PDE solvers can generally be divided into the following groups. The first group of work focuses on using neural networks with mesh-specific architecture design (such as convolutional layers for uniform mesh, or a graph layer for irregular mesh) to learn the spatial and/or temporal correlation of the PDE data [10, 38, 48, 64, 65, 75, 88, 91, 92, 100, 104, 109, 114, 116]. With input-target data collected, the training process can be conducted without the knowledge of underlying PDEs. This can be appealing when the physics of the system is unknown or partially known, such as large-scale climate modeling [61, 83, 90, 97]. The second group of work, namely the Physics-Informed Neural Networks (PINNs)[15, 40, 42, 52, 77, 87, 96, 106, 127], treat neural networks as a parametrization of the underlying solution function. PINNs incorporate the knowledge of the governing equations into the construction of loss function, which includes the residual of the PDE, the consistency with given boundary condition and initial condition. Unlike the previous group of works, PINNs do not necessarily need input-target data and can be trained solely based on equation loss. The third group of works, often referred to as the neural operator, focuses on learning a mapping between the function spaces[5, 8, 9, 16, 36, 43, 50, 54, 60, 68, 70, 71, 76, 78, 86]. Neural operator has the generalization capability within a family of PDE and can potentially be adapted to different discretization without retraining. DeepONet [76] proposes a practical realization of the universal operator approximation theorem [21]. Concurrent work graph neural operator [69] proposes a learnable kernel integral to approximate the solution operator of parametric PDEs and the follow-up work Fourier Neural Operator (FNO) [68] achieves excelling accuracy and efficiency on certain types of problems. Broadly speaking, the

---

[1]Code for this project is available at: `https://github.com/BaratiLab/FactFormer`.

operator learning can be conducted upon different types of function bases, such as the Fourier bases [34, 58, 59, 68, 95, 110, 121], wavelet bases [36], learned bases in an attention layer [16, 67], or based on approximation of the Green's function [7, 108]. The training of neural operators can also be combined with the principle of PINNs to yield a more physically consistent prediction [72, 117]. Our model is closely related to the neural operator, as the major building blocks in our proposed model are a learnable projection operator and a learnable kernel integral operator.

In addition to direct surrogate modeling, neural networks can also be combined with numerical solvers to improve their accuracy and efficiency. For example, using a trained neural network to correct the error of the solver on the fly [3, 28, 56, 89, 113], or doing offline high-fidelity reconstruction [24, 30, 49, 66, 102].

**Transformer for Physics Simulation** The Transformer model [115] have gained outstanding popularity in natural language modeling [13, 25], imagery data processing [27] and beyond [51]. In the field of physics simulation, Transformer has drawn increasing research interest as a surrogate model for simulation, with its modeling capability demonstrated both as a neural PDE solver [16, 32, 35, 41, 43, 48, 54, 67, 82, 86] and as a pure data-driven model in the absence of a known governing PDE [14, 20, 31, 83]. The dot-product attention can be considered as an approximation of an integral transform with a non-symmetric learnable kernel function [16, 17, 35, 54, 60, 122], which relates Transformer to other popular operator learning models such the FNO [68]. We will expand the discussion of Transformer under the kernel viewpoint in Section 3.

**Efficient Transformer** Following the introduction of Transformer [115], various works have investigated ways of reducing the computational cost of standard scaled-dot product attention. The first line of work seeks to remove the softmax and make use of matrix associativity to derive linear complexity attention [23, 53, 101], which has also been explored for PDE modeling[16, 43, 67]. The second line of work tries to approximate the dot product between query and key matrix by exploiting the low-rank structure of it [4, 22, 45, 55, 120, 123, 126]. Our work is related to the first group of works with a softmax-free design, but still calculates the dot product between query and key first. Among the second line of work, Axial Transformer is closely related to our work, as both works have explored conducting attention in an axial fashion. However, the derivation of attention matrix is different in the two works (see Section 3.2 for detailed comparison). More generally, the exploitation of the multi-dimensional tensor structure in our proposed model can be related to tensor factorization methods [57, 85] and their applications in various deep learning models [58, 63, 80, 84, 124].

## 3 Method

### 3.1 Attention mechanism

**Standard attention** Given three sets of vectors, namely the queries $\{\mathbf{q}_i\}_{i=1}^{N_q}$, keys $\{\mathbf{k}_i\}_{i=1}^{N_k}$, and values $\{\mathbf{v}_i\}_{i=1}^{N_v}$ (assuming $N_k = N_v$), attention mechanism [2, 33, 79, 115] dynamically computes a weighted average of the values: $\mathbf{z}_i = \sum_{j=1}^{N_v} h(\mathbf{q}_i, \mathbf{k}_j)\mathbf{v}_j$, where $\mathbf{q}_i, \mathbf{k}_i, \mathbf{v}_i \in \mathbb{R}^{1\times d}$, $h(\cdot)$ is the weight function that determines the contribution of a specific value to the final output. An example of $h(\cdot)$ is the scaled-dot product with softmax [115]: $h(\mathbf{q}_i, \mathbf{k}_j) = \exp\left(\mathbf{q}_i\mathbf{k}_j^T/\tau\right)/\sum_s \exp\left(\mathbf{q}_i\mathbf{k}_s^T/\tau\right)$, and $\tau$ is usually chosen as $\tau = \sqrt{d}$. The queries/keys/values are usually obtained from inputs via learnable projection. In self-attention, all of them are computed from the same source as follow:

$$\mathbf{q}_i = \mathbf{u}_i W_q, \mathbf{k}_i = \mathbf{u}_i W_k, \mathbf{v}_i = \mathbf{u}_i W_v, \tag{1}$$

where $\mathbf{u}_i \in \mathbb{R}^{1\times d_{\text{in}}}$ is the input vector and $\{W_q, W_k, W_v\} \in \mathbb{R}^{d_{\text{in}}\times d}$ are learnbale projection matrices. In cross-attention, queries are derived from one input while keys and values are derived from another.

**Attention as learnable integral** Under the hood of PDE modeling, the input sequence to the attention layer can be viewed as the sampling of input function on the discretization grid [16, 54, 60, 67]. Kovachki et al. [60] propose that the scaled-dot product attention [115] can be viewed as a special case of a Neural Operator [60], where the attention amounts to the Monte Carlo approximation of the learnable kernel integral. Cao [16] further proposes two interpretations of softmax-free attention. The first is to view attention as the Fredholm integral equation of the second kind with a learnable asymmetric dot-product kernel, and the second is to view it as a Peterov-Galerkin projection with learnable basis function. The softmax-free attention proposed by Cao [16] is later extended in OFormer [67], where Rotary Positional Encoding (RoPE) [105] is introduced to modulate the dot product and can be viewed as another special case of the kernel integral in Neural Operator style.

In this work, we continue on adopting the learnable kernel integral viewpoint of attention and view each channel of the hidden feature map as the sampling of a specific function on the discretization grid. Given query/key/value matrix $\{Q, K, V\} \in \mathbb{R}^{N \times d}$, their row vectors: $\mathbf{q}_i / \mathbf{k}_i / \mathbf{v}_i$, correspond to the sampling of a set of functions $\{q_l(\cdot), k_l(\cdot), v_l(\cdot)\}_{l=1}^d$ on grid point $x_i$, where $\{x_i\}_{i=1}^N$ discretizes the underlying domain. As a more concrete example, the $l$-th column (channel) of $\mathbf{q}_i$, represents the sampling of function $q_l(\cdot)$ on a grid point, i.e. $(\mathbf{q}_i)^l = q_l(x_i)$. Furthermore, softmax-free attention is equivalent to the numerical quadrature of a kernel integral:

$$(\mathbf{z}_i)^l = \sum_{s=1}^N w_s(\mathbf{q}_i \cdot \mathbf{k}_s)(\mathbf{v}_s)^l \approx \int_\Omega \kappa(x_i, \xi) v_l(\xi) d\xi, \tag{2}$$

where $\mathbf{z}_i$ is the output vector, $\kappa(x, \xi) = \sum_{l=1}^d q_l(x)k_l(\xi)$ is an instance-based kernel and $w_s$ is the quadrature weight. Understanding attention from the perspective of the kernel has been an active topic of research [17, 23, 111, 122]. The theoretical approximation power of different kernel integrals has also been analyzed under the context of PDE learning [35, 54, 60].

Note that the above kernel does not explicitly depend on the spatial coordinates $(x_i, \xi)$. For this work, we opt for a modified kernel formulation proposed in OFormer [67], which modulates the dot product kernel with relative position. Assuming the underlying spatial domain is 1-D (which is sufficient for our proposed model, see next subsection), given query and key vectors $\mathbf{q}_i, \mathbf{k}_j$ and their corresponding spatial coordinates $x_i, x_j$, RoPE [105] $(g(\cdot, \cdot) : \mathbb{R}^{1 \times d} \times \mathbb{R} \mapsto \mathbb{R}^{1 \times d})$ is defined as:

$$g(\mathbf{q}_i, x_i) = \mathbf{q}_i \boldsymbol{\Theta}(x_i), \quad g(\mathbf{k}_j, x_j) = \mathbf{k}_j \boldsymbol{\Theta}(x_j) \tag{3}$$

$$\text{where: } \boldsymbol{\Theta}(x_i) = \text{Diag}(R_1(x_i), \ldots, R_{d/2}(x_i)), \quad R_l = \begin{bmatrix} \cos(\lambda x_i \theta_l) & -\sin(\lambda x_i \theta_l) \\ \sin(\lambda x_i \theta_l) & \cos(\lambda x_i \theta_l) \end{bmatrix},$$

and $\lambda, \theta_l$ are hyperparameters. $\theta_l$ is usually chosen as $10000^{-2(l-1)/d}, l \in \{1, 2, \ldots, d/2\}$ following Vaswani et al. [115] and Su et al. [105]. $\lambda$ is a mesh-based weight that we heuristically set to $64$ throughout most problems. The projection function $\Theta(\cdot) : \mathbb{R} \mapsto \mathbb{R}^d \times \mathbb{R}^d$ can explicitly modulate the dot product with relative position: $g(\mathbf{q}_i, x_i) g(\mathbf{k}_j, x_j)^T = \mathbf{q}_i \Theta(x_i - x_j) \mathbf{k}_j^T$, thanks to the following property of rotation matrix: $R_l(x_i)R_l(x_j)^T = R_l(x_i - x_j)$.

To summarize, we will adopt attention mechanism in the following form for the proposed model (with modification discussed in the next subsection):

$$Z = w\tilde{Q}\tilde{K}^T V, \tag{4}$$

where $\tilde{\Box}$ denotes a matrix whose row vectors are RoPE encoded as in (3), e.g., $\tilde{Q}_i = g(\mathbf{q}_i, x_i)$, $w$ is the quadrature weight with a typical choice of $1/N$ for uniform quadrature rule, $Z$ is the output matrix. The query/key/value matrix $Q/K/V$ is derived from the input via learnable projections defined in (1). The matrix product $\tilde{Q}\tilde{K}^T$ evaluates the kernel function $\kappa(\cdot, \cdot)$ on the discretization grid $\{x_i\}_{i=1}^N$.

### 3.2 Multidimensional factorized attention

Compared to the standard scaled dot product attention that has quadratic complexity with respect to the length of the input sequence, the attention in (4) can enjoy a linear complexity by making use of the associativity of matrix multiplication (calculate $\tilde{K}^T V$ first). In PDE modeling, the length of the input sequence is equal to the number of points on the underlying discretization grid. Assuming the $n$-dimensional domain is discretized by $S_1 \times S_2 \times \ldots \times S_n = N$ points, the softmax-free attention in (4) will compute the kernel in (2) with the dot product of $Q$ and $K$, which are $N$ by $d$ matrices with $N$ usually much larger than $d$. The kernel matrix computed is by design low-rank as it is the product of two tall and thin matrices. Meanwhile, attending a large number of grid points to each other can be unstable and the linear attention has a complexity that is quadratic to the channel dimension $d$, which can limit the scalability of the model in terms of its width. To improve the numerical stability and reduce the computational cost of the aforementioned attention mechanism, we propose a simple yet efficient way to modify the kernel integral discussed in the previous section which is motivated by the low-rank structure of attention. Essentially, our model computes the kernel integral in an axial factorized manner instead of convolving over all the grid points in the domain.

For the following discussion, we will use tensor notation [57] to describe the operation. We assume the data is represented on a uniform Eulerian grid and can be treated as $n$-*way* tensor

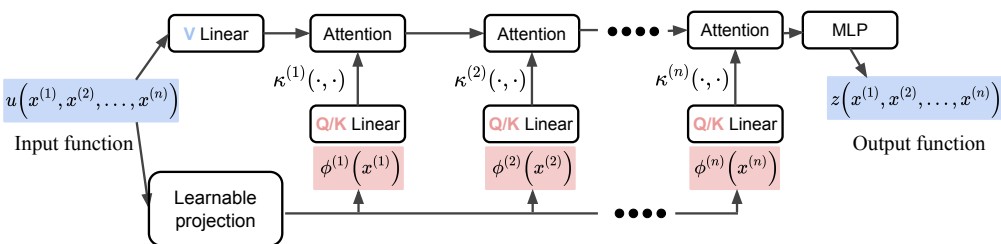

Figure 2: Schematic of the factorized kernel attention. **Upper path**: the input is transformed into the *Value* via a linear transformation. **Lower path**: the input is first projected into multiple sub-functions with a one-dimensional domain. These sub-functions are then used to derive the *Query* and *Key* on each axis, and their dot products form the kernel function of the corresponding axis. The *Value* is iteratively updated by the kernel integral transform along each axis and finally sent to an MLP.

$U \in \mathbb{R}^{S_1 \times S_2 \times \ldots \times S_n}$[2]. The product of it with a matrix $W \in \mathbb{R}^{J \times S_m}$ across the $m$-th mode will result in a tensor of shape $S_1 \times \ldots \times S_{m-1} \times J \times S_{m+1} \times \ldots \times S_n$, whose elements are defined as:

$$(U \times_m W)_{i_1 i_2 \ldots i_{m-1} j i_{m+1} \ldots i_n} = \sum_{i_m=1}^{S_m} U_{i_1 i_2 \ldots i_m \ldots i_n} W_{j i_m}. \tag{5}$$

**Learnable projection**   The first major component of the proposed framework is a set of learnable integral operators $\{\mathcal{G}^{(1)}, \mathcal{G}^{(2)}, \ldots, \mathcal{G}^{(n)}\}$ that projects the input function $u : \mathbb{R}^n \mapsto \mathbb{R}^d$ into a set of functions with one-dimensional domain $\{\phi^{(1)}, \phi^{(2)}, \ldots, \phi^{(n)}\} \in \mathbb{R} \mapsto \mathbb{R}^d$, which is defined as:

$$\phi^{(m)}(x_i^{(m)}) = \mathcal{G}^{(m)}(u)(x_i^{(m)}) \tag{6}$$
$$= h^{(m)} \left( w \int_{\Omega_1} \cdots \int_{\Omega_n} \gamma^{(m)} \left( u \left( \xi_1, \ldots, \xi_{m-1}, x_i^{(m)}, \xi_{m+1}, \ldots, \xi_n \right) \right) d\xi_1 \ldots d\xi_{m-1} d\xi_{m+1} \ldots d\xi_n \right),$$

where $h^{(m)}(\cdot) : \mathbb{R}^d \mapsto \mathbb{R}^d$ and $\gamma^{(m)}(\cdot) : \mathbb{R}^d \mapsto \mathbb{R}^d$ are pointwise learnable functions and $w = 1/(L_1 L_2 \cdots L_{m-1} L_{m+1} \cdots L_n)$, with $L_m$ being the size of domain $\Omega_m$ discretized by $\{x_i^{(m)}\}_{i=1}^{S_m}$. In practice, we implement $h^{(m)}$ as a three-layer multi-layer perception (MLP) similar to the feedforward network in Transformer [115] and $\gamma^{(m)}$ as a simple linear transformation. When the underlying grid is uniform, (6) simply amounts to first transforming the input with pointwise learnable functions $\gamma^{(m)}(\cdot)$, applying mean pooling over all but the $m$-th spatial dimension, and then applying another pointwise learnable function $h^{(m)}$.

**Factorized kernel integral**   Equipped with the above projection module, we now introduce our factorized kernel integral scheme. More specifically, we propose to use the following integral to replace the kernel integral in (2):

$$z \left( x_{i_1}^{(1)}, x_{i_2}^{(2)}, \ldots, x_{i_n}^{(n)} \right)$$
$$= \int_{\Omega_1} \kappa^{(1)}(x_{i_1}^{(1)}, \xi_1) \int_{\Omega_2} \kappa^{(2)}(x_{i_2}^{(2)}, \xi_2) \cdots \int_{\Omega_n} \kappa^{(n)}(x_{i_n}^{(n)}, \xi_n) v(\xi_1, \xi_2, \ldots, \xi_n) d\xi_1 d\xi_2 \ldots d\xi_n, \tag{7}$$

where kernels $\{\kappa^{(1)}, \ldots, \kappa^{(n)}\} : \mathbb{R} \times \mathbb{R} \mapsto \mathbb{R}$ are computed based on projected single-dimensional function along each axis, $v(\cdot) : \mathbb{R}^n \mapsto \mathbb{R}^d$ is derived from the input function $u$ via linear transformation (just as the value in standard attention). Next, we will discuss how the above kernel integral is implemented in practice. Using the learnable projection operator defined in (6), we can obtain $\{\hat{U}^{(1)}, \ldots, \hat{U}^{(n)}\}$ ($\hat{U}^{(m)} \in \mathbb{R}^{S_m \times d}$) from the input $U \in \mathbb{R}^{S_1 \times \ldots \times S_n \times d}$, where the $i_m$-th row of $\hat{U}$ is the evaluation of projected function $\phi^{(m)}(\cdot)$ at $x_{i_m}$: $\hat{U}_{i_m}^{(m)} = \phi^{(m)}(x_{i_m})$. Then we apply linear transformation on them to obtain the query/key matrix just as standard attention: $Q^{(m)} = \hat{U}^{(m)} W_q^{(m)}, K^{(m)} = \hat{U}^{(m)} W_k^{(m)}$, where $\{W_q^{(m)}, W_k^{(m)}\} \in \mathbb{R}^{d \times d}$ are learnable matrices. The query and key are used to compute the kernel matrix $A^{(m)} \in \mathbb{R}^{S_m \times S_m}$:

$$A^{(m)} = w_m \tilde{Q}^{(m)} \left( \tilde{K}^{(m)} \right)^T, \tag{8}$$

---

[2]In practice we often have an additional mode for the channel, resulting in a *(n + 1)-way* tensor.

where $w_m$ is the mesh weight, $\tilde{\square}$ denotes the RoPE encoded matrix as discussed in (4), the $i$-th row and $j$-th column of $A^{(m)}$ represents the kernel value $\kappa^{(m)}(x_i^{(m)}, x_j^{(m)})$. Despite (8) has a quadratic complexity with respect to the grid size $S_m$, this is an affordable cost for most of the problems where the axial grid size $S_m$ is mostly between 64 to 512. Meanwhile, the value $V \in \mathbb{R}^{S_1 \times \ldots \times S_n \times d}$ is derived from the input via a linear transformation (i.e. $(m + 1)$-th mode product): $V = U \times_{n+1} W_v$, where $W_v \in \mathbb{R}^{d \times d}$ is again a learnable matrix. The overall factorized kernel integral is numerically approximated with the following tensor-matrix product (Figure 2):

$$Z = \text{Att}(U) = V \times_1 A^{(1)} \times_2 A^{(2)} \times \ldots \times_n A^{(n)}. \tag{9}$$

In (9), the computation of all kernel is of complexity $O(S_1^2 d + S_2^2 d + \ldots + S_n^2 d)$, and the time complexity of a single tensor-matrix product $V \times_m A^{(m)}$ is $O(N S_m d)$. After evaluating the tensor-matrix product, the output tensor $Z$ will be sent to a pointwise feedforward network $f(\cdot) : \mathbb{R}^d \mapsto \mathbb{R}^d$. To sum up, the update protocol of a single layer in our proposed factorized Transformer is defined as follows:

$$U \leftarrow f\left(\text{IN}\left(\text{Att}\left(U\right)\right)\right) + U, \tag{10}$$

where $\text{Att}(\cdot)$ is the attention from (9), $\text{IN}(\cdot)$ is instance normalization [112] that normalizes each channel instance-wise.

It is worth pointing out that the axial factorized kernel proposed here shares some similarities with the Axial Transformer proposed in Ho et al. [45], but has two significant differences despite the connection. Firstly, Axial Transformer reduces the computational cost by constraining the context of attention along each axis (e.g. a pixel can only attend to other pixels on the same row), which amounts to moving all but one axis to the batch dimension. In this way computing the axial kernel matrix is of $O(N S_m d)$ complexity (recall $N = S_1 \times \ldots \times S_n$) instead of $O(S_m^2 d)$ as in our model. And its overall computation of attention is relatively more expensive due to the presence of softmax. Secondly, the decomposition in Axial Transformer is not layer-wise. For example, in the first layer, the attention is conducted in a row-wise manner and then the second block will conduct attention in a column-wise manner, whereas our model decomposes attention along all axes into a tensor-matrix product within every layer. We provide an illustrative example in Figure 27 of the Appendix.

### 3.3 Training techniques

In this subsection, we will discuss several techniques used for training the model (including baselines) in our numerical experiments. In general, these techniques aim to alleviate the compounding error of autoregressive neural PDE solvers when applied to time-dependent PDEs.

**Latent marching** It is proposed in the Li et al. [67] that a simple pointwise learnable function $\varepsilon(\cdot, \cdot) \in \mathbb{R}^d \times \mathbb{R}_{>0} \mapsto \mathbb{R}^d$ can be used to propagate dynamics in the latent space with a fixed time interval $\Delta t$: $z(x, t + \Delta t) = z(x, t) + \varepsilon(z(x, t), t)$, where $z$ is the output of the final attention layer. In practice, $\varepsilon$ is implemented as a pointwise MLP and is efficient to compute. Leveraging this technique, with one call to the neural solver, we can forward the state for multiple time steps (by marching in the latent space for $k$ steps), thus reducing the total number of calls by a ratio of $k$. This is in principle similar to the *Temporal Bundling* technique proposed in Message-Passing Neural PDE solver (MP-PDE) [12], yet different in practical realization. In MP-PDE, the multi-timestep prediction is implemented as first predicting the difference in time $\{d_1, d_2, \ldots, d_k\}$ and then adding them to the input $u_0$ by a forward Euler scheme in the physical space: $\hat{u}_k = u_0 + d_k \Delta t$. In this work, we opt for the latent marching to predict multi-timesteps as the forward Euler scheme (in the physical space) is less stable for fluid problems with relatively large time step sizes.

**Pushforward** Neural PDE solvers are observed to be unstable for time-dependent problems. A small error or perturbation that occurs at the beginning of neural PDE solvers' prediction, can easily result in an unbounded rollout error. While there is hardly a universal method for guaranteeing their stability, a wide array of techniques have been proposed to improve the stability of neural PDE solvers, such as adding physics constraints [72, 118, 119], rollout training [68] or adding random-walk noise [91, 100, 104]. For this work, we adopt the *pushforward* technique from MP-PDE, which amounts to rolling out the model for two steps during training and then letting the gradient only flows through the last step. This allows training the model on error-corrupted samples and promotes the stability of the model. From a practical perspective, this is straightforward to implement and also computationally much cheaper than standard rollout training.

# 4 Experiment

In this section, we will investigate our proposed model numerically on several challenging problems. Furthermore, we compare our model against softmax-free attention [16, 67]. The baseline models we compared against are Fourier Neural Operator (FNO) [68], Factorized Fourier Neural Operator (F-FNO) [110] and Dilated ResNet (Dil-ResNet) [44, 125]. FNO has been shown to have good accuracy on a wide range of PDE problems and is computationally very efficient owing to the Fast Fourier Transformation (FFT). F-FNO factorizes the spectral convolution in FNO into separate spectral convolution along different axes and adopt an improved residual connection formulation like Transformer [115]. Dil-ResNet is recently introduced by Stachenfeld et al. [104] to learn the coarse-grained dynamics of turbulent flow and has demonstrated state-of-the-art performance across several problems. We adopt the implementation of Dil-ResNet with group normalization from PDEArena [37]. On 2D steady-state problem where linear attention's computational cost is affordable, we also include the result from Galerkin Transformer [16], which uses CNN to project the function onto a coarse grid and applies linear attention on the coarse grid. The implementation details of the proposed model and baselines are available in Section A, B of the Appendix.

## 4.1 Benchmark problems

We first apply our model to three fluid-like systems, where the underlying physics patterns are sensitive to the spatiotemporal scale that discretization can resolve, and typically require fine discretization for classical numerical solvers. In these problems, the neural PDE solver is trained to predict the next frame (or multiple frames if using latent marching) given a context of previous frames. The number of context frames of Kolmogorov flow and isotropic turbulence is set to 10 following [68, 72], and 4 for smoke buoyancy similar to [37]. We also consider a well-known steady-state problem-2D Darcy flow, which has been studied in many of the previous works. Below we provide a brief description of each problem we studied. More details can be found in Section E of the Appendix.

**2D Kolmogorov flow**    The first example is 2D Kolmogorov flow governed by incompressible Navier-Stokes equation with a periodic boundary condition. The Reynolds number *Re* determines how turbulent the system will be. We adopt the setting of forced turbulence following Kochkov et al. [56] and generate the data by using the pseudo-spectral method to simulate fluid flow with Reynolds number $Re = 1000$. The objective is to predict the vorticity $\omega$ of the flow field within an interval $[t_0, t_0 + T]$, where $T = 1$s and $t_0$ is a random starting point in the sequence. We use a spatial grid of $256 \times 256$ and temporal discretization of $\Delta t = 0.0625$s (therefore 1s corresponds to 16 frames) to train and evaluate the model.

**3D isotropic turbulence**    The second example is 3D isotropic turbulence governed by incompressible Navier-Stokes equation with a periodic boundary condition. The major difference from the first example is that the vortex stretching term is non-zero for three-dimensional flow. We use the 3D spectral simulator from Mortensen and Langtangen [81], which simulates the forced turbulence described in Lamorgese et al. [62]. For generating the dataset, we simulate a system of Taylor Reynolds number $Re_\lambda = 84$ [62]. The objective is to predict the pressure $p$ and velocity $\mathbf{u}$ from $t = 0.5$ to $t = 1$s (10 frames). The model is trained and evaluated on a $60 \times 60 \times 60$ spatial grid with $\Delta t = 0.05$s.

**3D smoke buoyancy**    The third example is 3D buoyancy-driven flow, which depicts smoke volume rising in a closed domain. A similar system in 2D formulation has been studied in several previous works [11, 113]. The underlying governing equation is the incompressible Navier-Stokes equation coupled with an advection equation. The boundary condition for the smoke field is Dirichlet while the boundary condition for the flow field is Neumann. The advection equation describes the motion of smoke, which is transported along the flow field. We modify the solver from [37] that is implemented in *phiflow*[46] to generate the data, with buoyancy factor set to 0.5 and viscosity $\nu = 0.003$. The objective is to predict the scalar density field of smoke $d$ and velocity of flow $\mathbf{u}$ from $t = 3$ to $t = 15$s (16 frames). The model is trained and evaluated on a $64 \times 64 \times 64$ spatial grid with $\Delta t = 0.75$s. To account for non-periodic boundary conditions, we pad the domain for FNO variants and DilResNet following the original works. For FactFormer, we append a simple CNN block after the model, which comprises 3-by-3 convolutional layers with zero padding.

**2D Darcy flow**    In addition to the above time-depedendent systems, the fourth example is 2D steady-state problem from Li et al. [68]. Given the diffusion coefficient, the model predicts the steady-state flow field. The boundary condition is also Dirichlet so we adopt settings for all models similar to the 3D smoke problem.

## 4.2 Results and discussion

For all the models, we study two protocols of training. The first is Latent Marching with Pushforward (denote as **LM**). The second is simply Autoregressive (denote as **AR**), where the model is rolled out for two steps during training. For LM models, each call to the model will output $k$ future steps. On 2D Kolmogorov flow/3D smoke buoyancy, $k$ is set to $4$, and $2$ for 3D isotropic turbulence. We interleave pushforward training with standard per-step training for LM models. The relative $L^2$ norm is used to train and measure the error of each model following Li et al. [68]. The sequence-wise averaged error and the frame-wise error at the end frame are reported in Table 1, 2, 3. We also report the time cost of simulating a sequence and the number of parameters for each model. The frame-wise error trends are shown in Figure 6, 7, 8, 9, 10 in the Appendix. The visualization of predicted samples are provided in the Section F of Appendix.

We observe that Dil-ResNet has a slightly better per-frame fitting capability compared to the other models on 3D flow problems. As shown in the loss trend plots, it starts at a lower error compared to other models. This coincides with the observation in Stachenfeld et al. [104] where Dil-ResNet's performance is strong on 3D fluid problems. On 2D flow, F-FNO has the best accuracy compared to other models. Interestingly, FactFormer can catch up with Dil-ResNet on 2D Kolmogorov flow and 3D smoke buoyancy when the time duration becomes longer. Yet for shorter-term prediction - 3D isotropic turbulence, Dil-ResNet still has the best final accuracy. This suggests that the accuracy of long-term prediction can potentially benefit from exploiting the global structure that lies in the input. Nonetheless, compared to Dil-ResNet, FactFormer offers superior efficiency as indicated by the inference time (time cost of simulating a sequence). Since the training time is roughly proportional to the model forward time, on 3D problems Dil-ResNet generally takes 3-4 times longer to train. In terms of different training strategies, we find that AR models are less stable than multi-step training (LM) and computationally more expensive as it requires more calls to the neural solver. Despite the average error varies case by case, LM models' error generally accumulates slower on the problems we studied, whereas AR models quickly blow up in some cases.

Lastly, while Dil-ResNet has shown good accuracy for 3D flow problems, its performance is highly dependent on the training discretizations. As shown in the Figure 3, without changing model architecture, its evaluation errors increases significantly when the resolution increases, while Transformer-based models and FNO models' performance are roughly invariant to the resolution. This highlights a major difference between CNN-based models and neural operators.

| Model | FNO2D | | F-FNO2D | | Dil-ResNet | | FactFormer | |
|---|---|---|---|---|---|---|---|---|
| | AR* | LM | AR* | LM | AR | LM | AR | LM |
| $\omega$ avg. error | 0.3177 | 0.2978 | **0.1486** | 0.2453 | 0.8156 | 0.1655 | 0.8835 | 0.1734 |
| $\omega$ final error | 0.4423 | 0.4567 | **0.2811** | 0.3861 | 1.1692 | 0.3051 | 1.0963 | 0.3017 |
| Inf. time (s) | 0.73 | 0.81 | 0.86 | 1.01 | 4.69 | 1.78 | 3.14 | 1.38 |
| # params (M) | 85.1 | | 3.7 | | 2.4 | | 3.5 | |

Table 1: Evaluation results of 2D Kolmogorov flow. A batch size of 10 is used for inference. LM models predict 4 steps with each call to the model. Total prediction length is 16 steps. **AR***: Since for 2D problem FNO variants can afford to rollout more steps during training, AR FNO rollout for 12 steps, AR F-FNO rollout for 6 steps, whereas other AR models rollout for 2 steps during training. For model that has complex parameters, each `cfloat` parameter count as two paramaters.

| Model | FNO3D | | F-FNO3D | | Dil-ResNet | | FactFormer | |
|---|---|---|---|---|---|---|---|---|
| | AR | LM | AR | LM | AR | LM | AR | LM |
| $p$ avg. error | 0.8080 | 0.4634 | 0.3151 | 0.3264 | **0.1725** | 0.1778 | 0.2989 | 0.2545 |
| $p$ final error | 1.1285 | 0.6522 | 0.4250 | 0.4159 | 0.2573 | **0.2448** | 0.4407 | 0.3431 |
| **u** avg. error | 0.3967 | 0.3382 | 0.2298 | 0.2303 | **0.1143** | 0.1250 | 0.1775 | 0.1670 |
| **u** final error | 0.6561 | 0.4735 | 0.2799 | 0.2850 | 0.1675 | **0.1671** | 0.2594 | 0.2218 |
| Inf. time (s) | 1.01 | 0.91 | 2.77 | 1.37 | 12.67 | 6.89 | 2.68 | 1.31 |
| # params (M) | 509.8 | | 3.0 | | 6.9 | | 5.1 | |

Table 2: Evaluation results of 3D isotropic turbulence. A batch size of 4 is used for inference. LM models predict 2 steps with each call to the model. Total prediction length is 10 steps.

| Model | FNO3D | | F-FNO3D | | Dil-ResNet | | FactFormer | |
|---|---|---|---|---|---|---|---|---|
| | AR | LM | AR | LM | AR | LM | AR | LM |
| $d$ avg. error | 0.1607 | 0.1344 | 0.1038 | 0.1236 | **0.0843** | 0.0999 | 0.1017 | 0.0942 |
| $d$ final error | 0.1775 | 0.1287 | 0.1415 | 0.1219 | 0.1070 | 0.1062 | 0.1693 | **0.0941** |
| $\mathbf{u}$ avg. error | 0.5198 | 0.4255 | 0.3419 | 0.3713 | **0.2378** | 0.2747 | 0.3537 | 0.2592 |
| $\mathbf{u}$ final error | 1.0245 | 0.6718 | 0.8655 | 0.6146 | 0.5372 | 0.5023 | 0.7881 | **0.4482** |
| Inf. time (s) | 3.19 | 1.47 | 6.35 | 2.75 | 27.49 | 6.94 | 5.61 | 2.62 |
| # params (M) | 509.8 | | 3.0 | | 6.9 | | 4.6 | |

Table 3: Evaluation results of 3D smoke buoyancy. A batch size of 4 is used for inference. LM models predict 4 steps with each call to the model. Total prediction length is 16 steps.

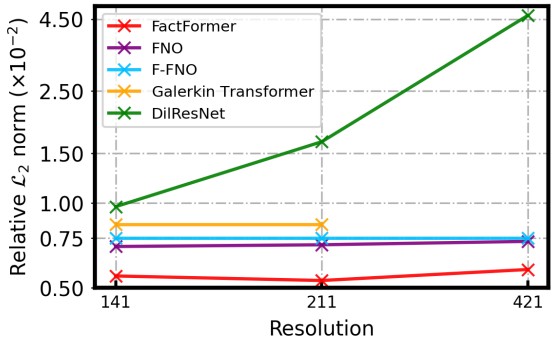

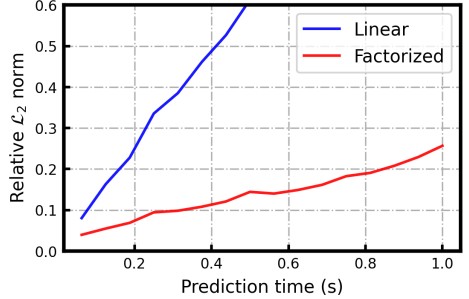

Figure 3: Error on 2D Darcy flow with different training resolutions. Galerkin Transformer's result is taken from the original paper [16].

Figure 4: Error trend of different Transformer models on 2D Kolmogorov flow with $128 \times 128$ grid.

| Model | Avg. rel. $L^2$ Norm | Fwd. | | Fwd. + Bwd. | |
|---|---|---|---|---|---|
| | | Enc. time (s) | Prop. time (s) | Time (s) | Mem. (MB) |
| Factorized attention | 0.1529 | 0.0202 | | 0.0954 | 5217 |
| Linear attention (`matmul`) | 0.5853 | 0.1370 | 0.0112 | 0.3013 | 12029 |
| Linear attention (`einsum`) | | 0.1333 | | 0.2938 | 12029 |

Table 4: Comparison between factorized and linear attention on their forward/backward computational cost, Mem. denotes the peak memory usage. The benchmark is carried out using PyTorch 1.8.2 on an RTX 3090, with a batch size of 4. **Enc. time**: the time spent on obtaining the latent encoding, primarily includes attention layers and feedforward layers after each attention layer; **Prop. time**: the time used to propagate dynamics in the latent space with a 3-layer MLP.

## 4.3 Comparison against full attention

In this subsection, we will present an ablation study of the proposed factorized attention mechanism with softmax-free attention (denoted as "linear attention") previously applied to PDE modeling[16, 67]. More specifically, we employ the attention from Li et al. [67] (in the form of (4)) to replace factorized attention in (10), with $\tilde{K}, V$ normalized column-wise via instance normalization, e.g. $||V_{\cdot,j}||_2 = 1$. To accommodate for the memory cost of linear attention, we further downsample the 2D Kolmogorov flow discussed in the last subsection to a $128 \times 128$ grid and train both linear and factorized attention models on it (with latent marching and pushforward trick).

**Comparison of performance** We compare the accuracy and computational cost of the two attention mechanisms in Figure 4 and Table 4. While in principle full attention could have better approximation capacity than factorized attention, in practice we find that it performs worse than factorized attention on this problem we studied. Specifically, its rollout is less stable and results in a degraded accuracy. We hypothesize that this is due to the instability of iteratively calculating the attention matrix of a large size, as rolling out the prediction requires recursively calling the model multiple times. In

addition to the accuracy improvement, the benchmark on computation empirically demonstrates the computational efficiency improvement of factorized attention over linear attention. We provide more detailed comparison between factorized attention and linear attention in the Section D of Appendix, where we observed consistent efficiency improvement with different grid sizes and model sizes.

**Pattern of attention matrices**   We also investigate the structure of different attention matrices. By construction, when using softmax-free attention to compute the kernel integral in (2), the kernel matrix $A = QK^T$ is going to have a low-rank structure since $\text{rank}(A) \leq \min(\text{rank}(Q), \text{rank}(K))$, $Q, K \in \mathbb{R}^{N \times d}$ and usually $N >> d$. After training, we compute the attention matrices based on 100 samples and conduct singular value decomposition (SVD) on them. We define the total energy of the spectrum as the sum of singular values $E = \sum_i \sigma_i$, where $\sigma_i$ is the $i$-th singular value and report the normalized cumulative energy histogram $b_k = \sum_{i=1}^{k} \sigma_i / \sum_i \sigma_i$ in Figure 5a, 5b, 5c. For each layer, the histogram is averaged across the attention matrices of all heads. We observe that for linear attention, its rank is relatively low as less than 5% of the singular values capture over 90% of the total energy, which is similar to the trend observed from previous works studying the rank of standard softmax-attention [6, 26, 120]. Note that the spectrum of linear attention is based on a truncated SVD and therefore its rank will be even lower if a full SVD is performed. The highly low-rank structure of the full attention matrix hints the potential to approximate with or decomposed into smaller and more compact matrices, and our proposed factorized scheme is one example.

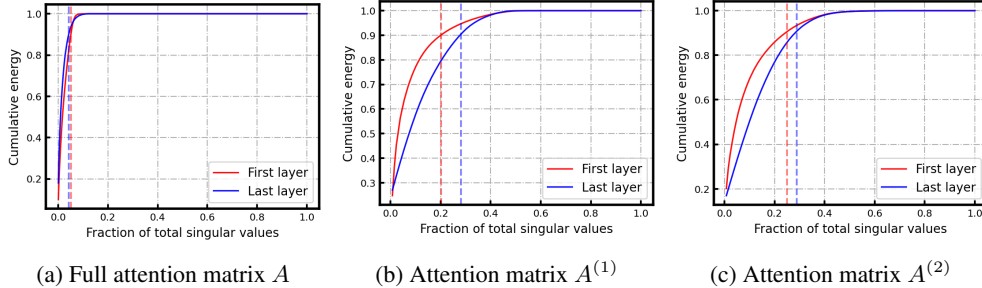

|  (a) Full attention matrix $A$ | (b) Attention matrix $A^{(1)}$ | (c) Attention matrix $A^{(2)}$ |

Figure 5: Spectrum of different attention matrices. **(a)**: Attention matrices from softmax-free attention. **(b), (c)**: Attention matrices for axis $x$ and $y$ from FactFormer. The vertical line indicates the fraction of singular values that capture 90% of the total energy. Since the full attention matrix $A$ has a relatively large size ($16384 \times 16384$). Therefore its spectrum is computed via TruncatedSVD [39] with top 1024 components truncated.

## 5   Conclusion

In this work, we propose an end-to-end Transformer for PDE modeling, which features a learnable projection operator and a factorized kernel integral. We demonstrate that the proposed model balances efficiency and accuracy well, making it a promising and scalable solution for PDE surrogate modeling. However, the proposed attention mechanism is still not free from the curse of dimensionality. The computation of the factorized kernel integral requires evaluating the function on all $S_1 \times S_2 \times \ldots \times S_m$ grid points. A future direction could be extending the factorization scheme to a more efficient tensor decomposition format like tensor-train. The proposed model currently exploits the uniform structure of the underlying grids and use mean pooling when doing projection, but non-uniform quadrature weight will be necessary when applying to non-uniform grids. It is also observed that the proposed model and other neural PDE solvers can be unstable due to the error accumulation when solving time-dependent systems.

## Acknowledgement

This work is supported by the National Science Foundation under Grant No. 1953222. The authors would like to thank the annonymous reviewers and area chair for their efforts and valuable feedback during the reviewing process. The authors would like to thank Dr. Shuhao Cao from University of Missouri - Kansas City for the comments regarding the backpropagation of feedforward layer and attention layer in Transformer. The authors would also like to thank Zhi Ye for the suggestions on benchmarking the computational cost of the model.

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

# A Model implementation details

The major hyperparameters are listed in Table 5.

| Hyperparameter | 2D Kolmogorov | 3D turbulence | 3D smoke | 2D Darcy flow |
|---|---|---|---|---|
| Hidden dimension | 128 | 128 | 128 | 128 |
| Depth | 4 | 4 | 3 | 3 |
| Heads | 8 | 6 | 6 | 12 |
| Kernel dimension | 128 | 192 | 192 | 128 |
| Input encoder | 2D Conv | 2D Conv | 2D Conv | MLP |
| Output decoder | MLP | MLP | MLP | MLP |

Table 5: Major hyperparameters for FactFormer

*Hidden dimension* indicates the number of channels in the latent space. *Depth* denotes the number of attention layers. Before entering each attention layer, we modulate the latent encoding with positional encoding: $z_i \leftarrow z_i + \psi(x_i)$, where $x_i$ is the Cartesian coordinate of latent encoding $z_i$, $\psi : \mathbb{R}^n \mapsto \mathbb{R}^d$ is a random Fourier feature mapping [94, 107] with a learnable linear transformation. *Kernel dimension* is the dimension of each head, which is equivalent to $d_k$, the number of function bases used to compute the kernel: $\kappa(x, \xi) = \sum_l^{d_k} q_l(x) k_l(\xi)$. We train the model with AdamW optimizer and cyclic learning rate scheduler with a maximum learning rate $3e - 4$, similar to prior Transformer-for-PDE works [16, 43, 67].

For 3D smoke buoyancy and 2D Darcy flow problem, we append a CNN block after attention layers to better account for the boundary values. The CNN block has a U-shape arrangement with 4 CNN layers, with all layers using a kernel size of 3 and padding size of 1 (pad with zeros). The first CNN layer has a stride of 2, while other layers have a stride of 1. The stride-2 convolution will downsample the data by half, so nearest upsampling is applied between the second and third CNN layers to recover the spatial resolution.

On top of every model (including baselines we will discuss in the next section), we use a 2D convolutional layer to compress the temporal dimension if it is a time-dependent problem. Concretely, we first reshape the input into $(N, T_{\text{in}})$ where $N$ is the number of spatial grid points and $T_{\text{in}}$ is the number of input frames. Then we apply 2D convolution filters of size $(1, T_{\text{in}})$ to compress the temporal dimension to 1. At the bottom of every model, we use a three-layer MLP to project the latent encoding back to variables of interest such as pressure and velocities. In addition, we adopt a curriculum training strategy for all latent marching models, where we only march for 1 step at the beginning of the training and don't do any pushforward. Then we gradually increase the latent marching steps throughout the training and apply pushforward when the model has been trained for around 6% of the total epochs.

# B Baseline implementation details

In this section, we provide the full details of baseline models, namely FNO, F-FNO, Dil-ResNet, and linear attention Transformer.

For Fourier Neural Operator, the implementation is taken from Li et al. [72]'s official implementation: https://github.com/neuraloperator/physics_informed. And for Factorized Fourier Neural Operator the implementation is taken from https://github.com/alasdairtran/fourierflow. We add group normalization before the final fully-connected layer. We use a hidden size of 96, a mode number of 12 for 3D problems, 24 for 2D turbulence, 20 for Darcy flow, and a layer number of 4.

For Dil-ResNet, we adopt the implementation from Gupta and Brandstetter [37]:https://github.com/microsoft/pdearena. Compared to Gupta and Brandstetter [37] and Stachenfeld et al. [104], we simplify the setting by truncating the number of layers inside each block, where we use dilation layers $[1, 3, 8, 3, 1]$ ($[1, 2, 4, 2, 1]$ is chosen for the 3D problem as it has slightly better performance) inside each block instead of $[1, 2, 4, 8, 4, 2, 1]$. The primary reason is that without truncation, the training on 3D problems will take over a week (and cannot fit into a single A6000 GPU for 2-step

rollout training), which is significantly slower than other models. In summary, we use 3 residual blocks (each with 5 CNN layers of width 128), an MLP-based or 2D convolution-based (for convolution in temporal domain) encoder/decoder, with group normalization inserted between every residual block. The implementation of the linear attention Transformer follows OFormer's [67] attention implementation https://github.com/BaratiLab/OFormer, with a Galerkin style normalization scheme [16]. Other hyperparameters are kept the same as the hyperparameters listed in Table 5 - 2D Kolmogorov. For non-Transformer models, they are trained with Adam optimizer and decay learning rate from $5e - 4$ to $5e - 6$ throughout the training.

All the experiments are carried out using PyTorch 1.8 except for FNO/F-FNO experiment, which uses PyTorch 1.13 for optimizing complex-valued parameters. We train all LM models for 100k iterations and AR models for 64k iterations of gradient updates.

## C   Visualization of error trend

This section includes the average frame-wise error trend for the time-dependent systems we have investigated.

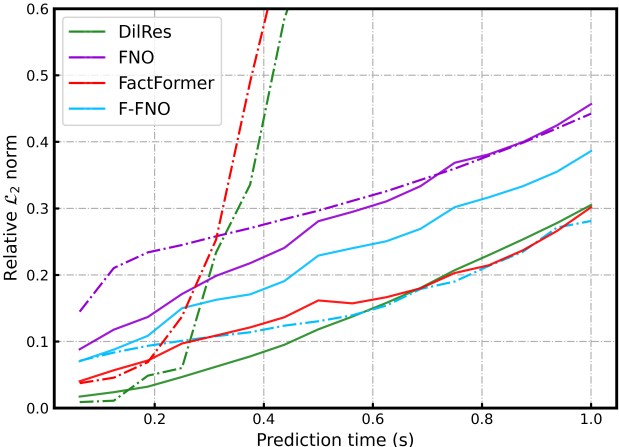

Figure 6: Error trend of vorticity $\omega$ on 2D Kolmogorov flow.
**Dashed line**: AR; **Solid line**: LM

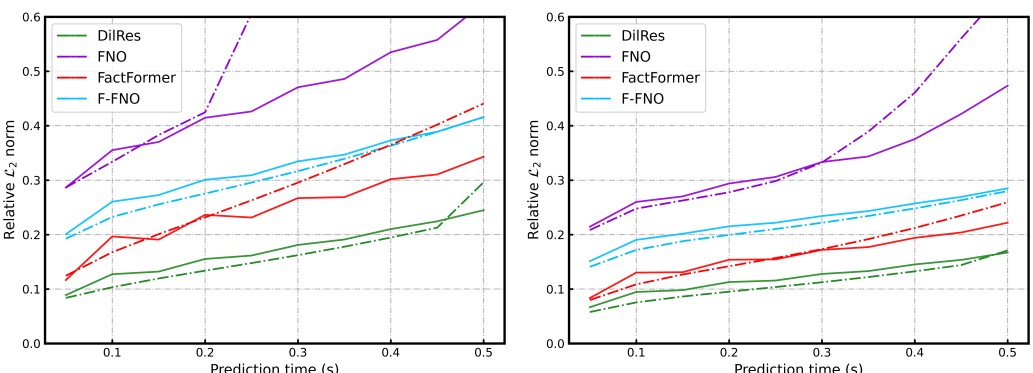

Figure 7: Error trend of pressure $p$ on 3D isotropic turbulence.

Figure 8: Error trend of velocity $\mathbf{u}$ on 3D isotropic turbulence.

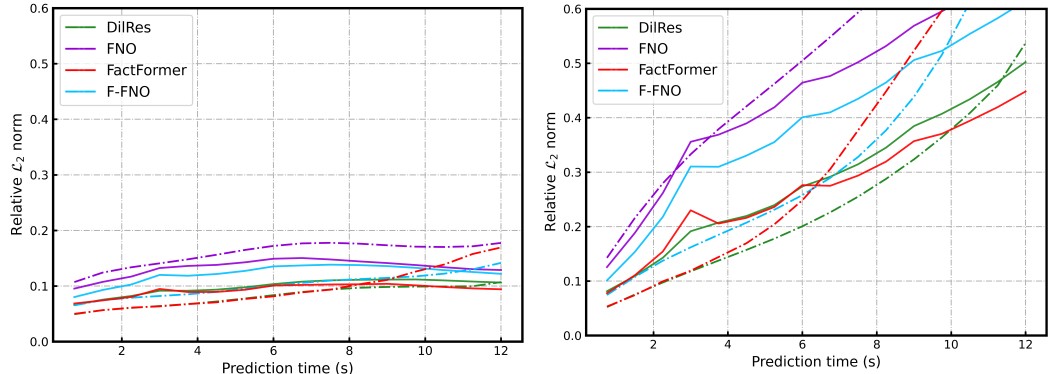

Figure 9: Error trend of marker field $d$ on 3D smoke buoyancy.

Figure 10: Error trend of velocity field $\mathbf{u}$ on 3D smoke buoyancy.

## D Further ablation study

This section includes further study and numerical experiments on the proposed model.

**Note on backpropagation** Consider a single (softmax-free) attention block that consists of a attention layer and a two layer feedforward network (for simplicity we consider single-headed case):

$$U_1 = \text{Att}(U_0), \quad U_2 = \sigma(U_1 W_1) W_2, \tag{11}$$

where $U_0 \in \mathbb{R}^{N \times d}$ is the input, $W_1, W_2 \in \mathbb{R}^{d \times d}$ are learnable weights for the feedforward network. Given a loss function $l(\cdot) : \mathbb{R}^{N \times d} \mapsto \mathbb{R}$ (for example, mean squared error), define $\tilde{U}_1 := U_1 W_1$, $\frac{\partial l}{\partial U_2} := h$, $\frac{\partial \sigma(\tilde{U}_1)}{\partial \tilde{U}_1} := g$, the gradient for the weights and input in the feedforward network are ($\odot$ denotes element-wise multiplication):

$$\frac{\partial l}{\partial W_2} = \sigma(\tilde{U}_1)^T h \tag{12}$$

$$\frac{\partial l}{\partial W_1} = U_1^T (h W_2^T \odot g) \tag{13}$$

$$\frac{\partial l}{\partial U_1} = (h W_2^T \odot g) W_1^T \tag{14}$$

For linear (softmax-free) dot-product attention: $U_1 = \text{Att}(U_0) = Q K^T V$, where $Q = U_0 W_q$, $K = U_0 W_k$, $V = U_0 W_v$, and thus $U_1 = U_0 W_q W_k^T U_0^T U_0 W_v$, the gradient of weights are:

$$\frac{\partial l}{\partial W_q} = U_0^T \frac{\partial l}{\partial U_1} W_v^T U_0^T U_0 W_k, \tag{15}$$

$$\frac{\partial l}{\partial W_k} = U_0^T U_0 W_v (\frac{\partial l}{\partial U_1})^T U_0 W_q, \tag{16}$$

$$\frac{\partial l}{\partial W_v} = U_0^T U_0 W_k W_q^T U_0^T \frac{\partial l}{\partial U_1}, \tag{17}$$

where $U_0 \in \mathbb{R}^{N \times d}$ has appeared three times in each calculation and thus in backpropagation the computational cost of attention layer is generally more expensive than the feedforward layer where it involves more matrices that grow exponentially with respect to the spatial resolution.

Next we provide a comparison between the backpropagation of linear attention and the proposed factorized attention in scalar summation form. For simplicity, we only consider the attention layer. For linear attention, it can be written as:

$$Z_{i,c} = \sum_{m=1}^{d} Q_{i,m} (\sum_{j=1}^{N} K_{j,m} V_{j,c}), \tag{18}$$

where $Z_{i,c}$ denotes the $i$-th row and $j$-th column of matrix $Z$ and similar for other matrices, $Q = XW_q, K = XW_k, V = XW_v$ and $X \in \mathbb{R}^{N \times d}$ is input. The gradient of parameters are computed as:

$$\frac{\partial Z_{i,c}}{\partial (W_q)_{r,s}} = X_{i,r}(\sum_{j=1}^{N} K_{j,s} V_{j,c}), \tag{19}$$

$$\frac{\partial Z_{i,c}}{\partial (W_k)_{r,s}} = Q_{i,s}(\sum_{j=1}^{N} X_{j,r} V_{j,c}), \tag{20}$$

$$\frac{\partial Z_{i,c}}{\partial (W_v)_{r,s}} = \begin{cases} \sum_{m=1}^{d} Q_{i,m}(\sum_{j=1}^{N} K_{j,m} X_{i,r}) & : \text{if } s = c \\ 0 & : \text{otherwise.} \end{cases} \tag{21}$$

For the proposed factorized attention, it can written as:

$$Z_{i,c} = \sum_{j_1=1}^{S_1} \sum_{j_2=1}^{S_2} \cdots \sum_{j_n=1}^{S_n} A_{i_1,j_1}^{(1)} A_{i_2,j_2}^{(2)} \cdots A_{i_n,j_n}^{(n)} V_{j,c}, \quad j := (j_1, j_2, \ldots, j_n), \ i := i_1, i_2, \ldots, i_n$$
$$\tag{22}$$

where $A^{(m)} = Q^{(n)}(K^{(m)})^T$ ($A^{(m)} \in \mathbb{R}^{S_m \times S_m}, N = S_1 \times S_2 \times \ldots \times S_n$) is the axial kernel matrix as defined in (8), $X^{(m)} \in \mathbb{R}^{S_m \times d}$ is the axial projection along $m$-th axis as defined in (6).

For $W_v$, its gradient is computed as:

$$\frac{\partial Z_{i,c}}{\partial (W_v)_{r,s}} = \begin{cases} \sum_{j_1=1}^{S_1} \sum_{j_2=1}^{S_2} \cdots \sum_{j_n=1}^{S_n} A_{i_1,j_1}^{(1)} A_{i_2,j_2}^{(2)} \cdots A_{i_n,j_n}^{(n)} X_{j,r} & : \text{if } s = c \\ 0 & : \text{otherwise.} \end{cases} \tag{23}$$

For $W_q^{(n)}, W_k^{(n)}$, their gradients are computed as:

$$\frac{\partial Z_{i,c}}{\partial (W_q^{(n)})_{r,s}} = X_{i_n,r}^{(n)} \sum_{j_1=1}^{S_1} \sum_{j_2=1}^{S_2} \cdots \sum_{j_n=1}^{S_n} A_{i_1,j_1}^{(1)} A_{i_2,j_2}^{(2)} \cdots A_{i_{n-1},j_{n-1}}^{(n-1)} V_{j,c} K_{j_n,s}^{(n)}, \tag{24}$$

$$\frac{\partial Z_{i,c}}{\partial (W_k^{(n)})_{r,s}} = Q_{i_n,s}^{(n)} \sum_{j_1=1}^{S_1} \sum_{j_2=1}^{S_2} \cdots \sum_{j_n=1}^{S_n} A_{i_1,j_1}^{(1)} A_{i_2,j_2}^{(2)} \cdots A_{i_{n-1},j_{n-1}}^{(n-1)} V_{j,c} X_{j_n,r}^{(n)}, \tag{25}$$

The major difference is that for factorized attention the summation is taken over each axis separately while for linear attention is taken over all $N$ grid points.

**Runtime comparison** As discussed in Section A, the kernel dimension indicates how many function bases are used to evaluate the kernel and a larger kernel dimension is beneficial to the learning capacity of the model. As shown in Figure 11a, 11b, linear attention's training cost increases more significantly than the factorized attention as kernel dimension increases, since its complexity is quadratic with respect to kernel dimension. Factorized attention's computational efficiency can be further improved by reducing the spatial resolution, leveraging techniques such as learning the mapping in the latent space (similar to latent diffusion model [99]), multi-scale network architecture that resembles multigrid methods [35, 74], or domain decomposition [47, 93]. Furthermore, the training cost of factorized attention is also relatively lower than linear attention on 3D domain as shown in Figure 12a, 12b.

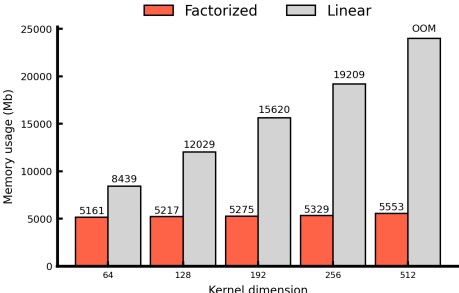

(a) Forward + backward peak memory usage.

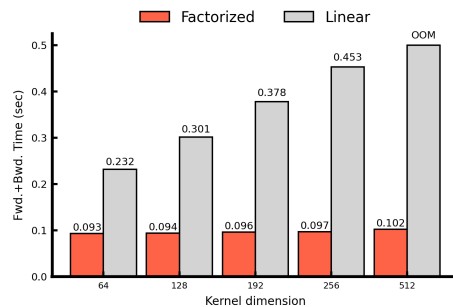

(b) Forward + backward time per iteration.

Figure 11: Benchmark of factorized attention and linear attention on 2D domain (with $128 \times 128$ grid) with varying kernel dimension. Benchmark is done on an RTX 3090 with PyTorch 1.8.2 and a batch size of 4. Hyperparameter setting is the same as in Table 5-2D Kolmogorov flow. "OOM" denotes out of memory.

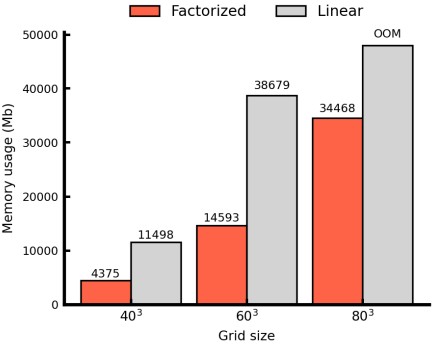

(a) Forward + backward peak memory usage.

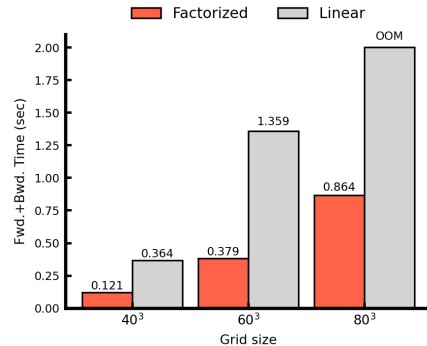

(b) Forward + backward time per iteration.

Figure 12: Benchmark of factorized attention and linear attention on 3D domain with varying grid size. Benchmark is done on an A6000 with PyTorch 1.8.2 and batch size of 1. Hyperparameter setting is the same as in Table 5-3D isotropic turbulence.

**Model scaling performance**   We study the impact of the number of heads and size of kernel dimension on prediction loss. For each direction of hyperparameter search, we fix the value of other hyperparameters to that shown in Table 5. The ablation experiments are conducted on 2D Kolmogorov flow (sampled from a $128 \times 128$ grid) with a splitting different from the Evaluation Section in the main body of the paper. As shown in Figure 13b, the number of attention heads has a crucial impact on the final performance. The model's performance drops significantly when using fewer heads. This highlights the importance of multi-head mechanism in the factorized attention. In addition, we observe that the final accuracy of our model benefits from an increased kernel dimension (as shown in Figure 13a).

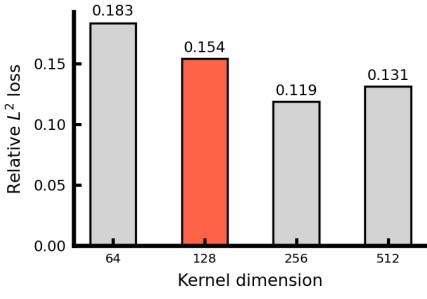

(a) Ablation on the kernel dimension.

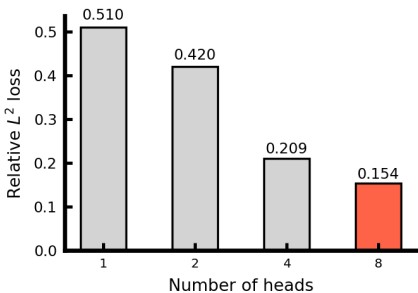

(b) Ablation on the number of heads.

Figure 13: Ablation study on the key hyperparameters. Red color denotes the final choice of hyperparameter. Experiments are carried out on the validation fold.

**Visualization of learned kernels**   We visualize the learned kernel as shown in Figure 14. Due to the presence of Rotary positional encoding [105], all kernels have a stationary pattern (the kernel value $\kappa(\xi_1, \xi_2)$ depends only on the relative distance between two points, e.g. $L^2$ distance: $\|\xi_1 - \xi_2\|_2$). The kernel matrices also exhibits symmetric pattern despite the non-symmetric nature of dot product $QK^T$ and all kernel matrices are diagonal dominated.

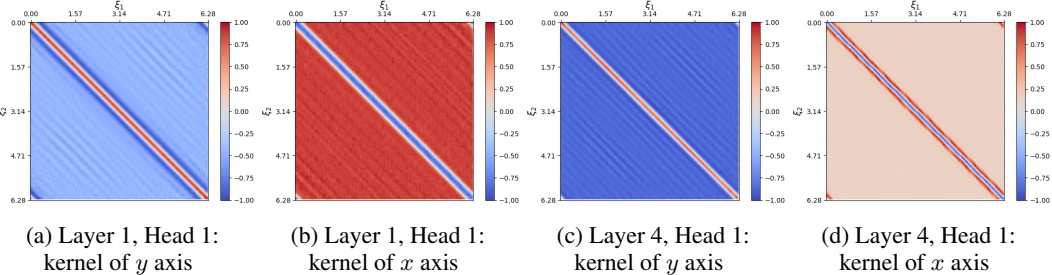

(a) Layer 1, Head 1: kernel of $y$ axis

(b) Layer 1, Head 1: kernel of $x$ axis

(c) Layer 4, Head 1: kernel of $y$ axis

(d) Layer 4, Head 1: kernel of $x$ axis

Figure 14: Visualization of normalized attention kernel .

**Influence of random seed**   We investigate the influence of random seeds by training the model with three different seeds. As shown in Figure 15, all models converge to the similar level of loss with marginal difference.

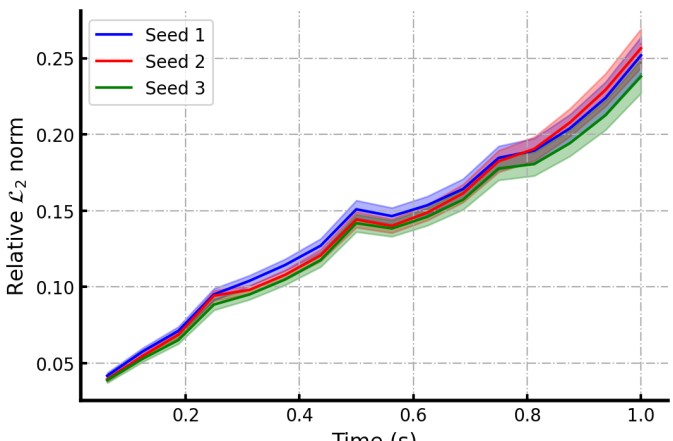

Figure 15: Averaged frame-wise loss trends. Each loss curve corresponds to a model initialized under a specific seed.

## E   Dataset details

In this section we provide the details for each dataset.

**2D Kolmogorov flow**   The incompressible Navier-Stokes equation under vorticity form reads as,

$$\frac{\partial \omega(\mathbf{x}, t)}{\partial t} + \mathbf{u}(\mathbf{x}, t) \cdot \nabla \omega(\mathbf{x}, t) = \frac{1}{Re} \nabla^2 \omega(\mathbf{x}, t) + f(\mathbf{x}), \qquad \mathbf{x} \in (0, 2\pi)^2, t \in (0, T],$$

$$\nabla \cdot \mathbf{u}(\mathbf{x}, t) = 0, \qquad \mathbf{x} \in (0, 2\pi)^2, t \in [0, T], \quad (26)$$

$$\omega(\mathbf{x}, 0) = \omega_0(\mathbf{x}), \qquad \mathbf{x} \in (0, 2\pi)^2,$$

where $\omega$ denotes vorticity, $\mathbf{u}$ denotes velocity, *Re* denotes the Reynolds number, $\mathbf{x} = (x_1, x_2)$ denotes the spatial coordinates and $f(\cdot)$ is the forcing term that is set to $f(\mathbf{x}) = -n \cos(n x_2) - 0.1\omega(\mathbf{x})$. The equation is periodic in all spatial directions. Compared to the cases discussed in Li et al. [72], we set the forcing factor $n$ to 8 [1, 19] and introduce dragging force term $0.1\omega(\mathbf{x})$ as described in Kochkov et al. [56]. The initial condition $\omega_0$ is sampled from a prescribed Gaussian random field

same as Li et al. [72]. The dataset consists of 100 trajectories for training and 20 trajectories for testing, with the length of each trajectory being 10 seconds and 160 frames.

We modify the pseudo-spectral solver (under Apache License 2.0) from https://github.com/neuraloperator/physics_informed/blob/master/solver/kolmogorov_flow.py to generate the data. The referenced direct numerical simulation is carried out with a spatial resolution of $2048 \times 2048$ and a temporal resolution of $1e-4$.

**3D isotropic turbulence**   The incompressible Navier-Stokes equation for this problem is given as:

$$\frac{\partial \mathbf{u}(\mathbf{x}, t)}{\partial t} + \mathbf{u}(\mathbf{x}, t) \cdot \nabla \mathbf{u}(\mathbf{x}, t) = \nu \nabla^2 \mathbf{u}(\mathbf{x}, t) - \frac{1}{\rho} \nabla p(\mathbf{x}, t) + \mathbf{f}(\mathbf{x}), \quad \mathbf{x} \in (0, 2\pi)^3, t \in (0, T],$$

$$\nabla \cdot \mathbf{u}(\mathbf{x}, t) = 0, \qquad\qquad\qquad \mathbf{x} \in (0, 2\pi)^3, t \in [0, T], \quad (27)$$

$$\mathbf{u}(\mathbf{x}, 0) = \mathbf{u}_0(\mathbf{x}), \qquad\qquad\qquad \mathbf{x} \in (0, 2\pi)^3,$$

where $\mathbf{u}$ denotes velocity, $p$ denotes the pressure, $\nu$ is the viscosity parameter, $\mathbf{x} = [x_1, x_2, x_3]$ denotes the spatial coordinates and $f(\cdot)$ is the forcing term. The equation is periodic in all three spatial dimensions. The initialization of $\mathbf{u}$ and the forcing settings follow Rogallo [98] and Lamorgese et al. [62] respectively, with Taylor Reynolds number set to $84$ [62]. The dataset consists of 1000 trajectories for training and 100 trajectories for testing, with the length of each trajectory being 1 second and 20 frames.

We use the spectral Galerkin solver (under GNU GPL license 3.0) from https://github.com/spectralDNS. The referenced simulation is carried out with a spatial resolution of $60 \times 60 \times 60$ and a temporal resolution of $0.005s$.

**3D smoke buoyancy**   The governing equations for the 3D smoke buoyancy problem are incompressible Navier-Stokes equation (similar as above) coupled with advection equation:

$$\frac{\partial \mathbf{u}(\mathbf{x}, t)}{\partial t} + \mathbf{u}(\mathbf{x}, t) \cdot \nabla \mathbf{u}(\mathbf{x}, t) = \nu \nabla^2 \mathbf{u}(\mathbf{x}, t) - \frac{1}{\rho} \nabla p(\mathbf{x}, t) + \mathbf{f}(\mathbf{x}, t), \quad \mathbf{x} \in (0, L)^3, t \in (0, T],$$

$$\frac{\partial d(\mathbf{x}, t)}{\partial t} + \mathbf{u}(\mathbf{x}, t) \cdot \nabla d(\mathbf{x}, t) = 0, \qquad\qquad \mathbf{x} \in (0, L)^3, t \in (0, T],$$

$$\nabla \cdot \mathbf{u}(\mathbf{x}, t) = 0, \qquad\qquad\qquad \mathbf{x} \in (0, L)^3, t \in [0, T],$$

$$\mathbf{u}(\mathbf{x}, 0) = 0, \quad \mathbf{d}(\mathbf{x}, 0) = d_0(\mathbf{x}), \qquad\qquad \mathbf{x} \in (0, L)^3,$$

where $\mathbf{f}(\mathbf{x}, t) = [0, 0, \eta d(\mathbf{x}, t)]$, $\eta$ is the buoyancy factor, the velocity field $\mathbf{u}$ has a Dirichlet boundary condition: $\mathbf{u}(\mathbf{x}, \cdot) = 0, \forall \mathbf{x} \in \partial \Omega$, and the scalar density field for smoke has a Neumann boundary condition: $\nabla d(\mathbf{x}, \cdot) = 0, \forall \mathbf{x} \in \partial \Omega$. The initial condition $d_0(\mathbf{x})$ is a random field [3] with scaling of Fourier coefficient set to 15.0, smoothness factor set to 4.0. The length of the rectangular domain $L$ is set to 8. The dataset consists of 2000 trajectories for training and 200 trajectories for testing, with the length of each trajectory being 15 seconds and 20 frames.

We modify the 2D solver (under MIT license) from https://github.com/microsoft/pdearena to generate the data. The solver applies an advection-project scheme. The referenced simulation is carried out with a spatial resolution of $64 \times 64 \times 64$ and a temporal resolution of $0.75s$.

**2D Darcy flow**   The equation for the 2D Darcy flow is defined as:

$$-\nabla \cdot (a(x) \nabla u(x)) = f(x), \quad x \in (0, 1)^2,$$
$$u_0(x) = 0, \quad x \in \partial(0, 1)^2, \tag{28}$$

where $f(x)$ is the forcing function that is set to constant 1. The coefficient function $a(x)$ is sampled from Gaussian Random Field with zero Neumann boundary condition. The data is generated via second-order finite difference solver on a $421 \times 421$ resolution grid. We use the pre-generated dataset from Li et al. [68] (under MIT license). The dataset consists of 1000 samples for training and 100 samples for testing.

---

[3]Implemented with *phiflow*'s Noise class, see:https://tum-pbs.github.io/PhiFlow/phi/field/

# F   Results visualization

In this section, we provide exemplary visualization of the model's prediction. For 3D problems, the cross-section at the middle of the first axis is shown.

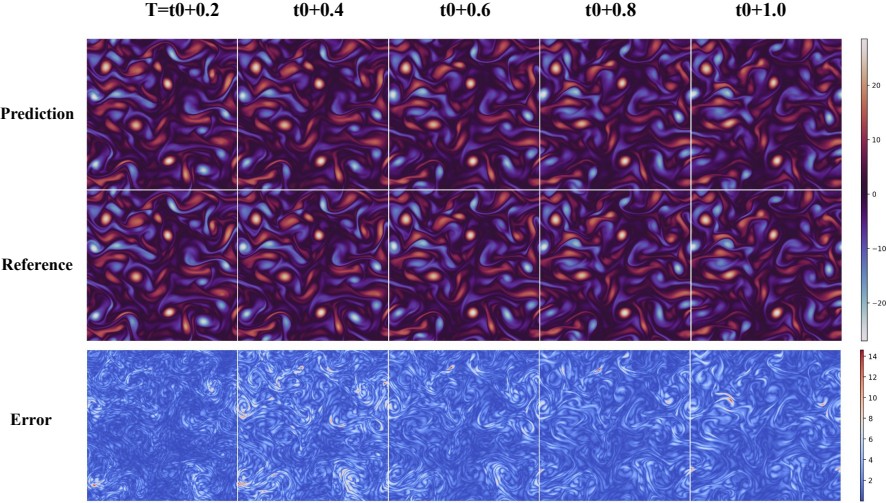

Figure 16: Sample 1 of 2D Kolmogorov flow.

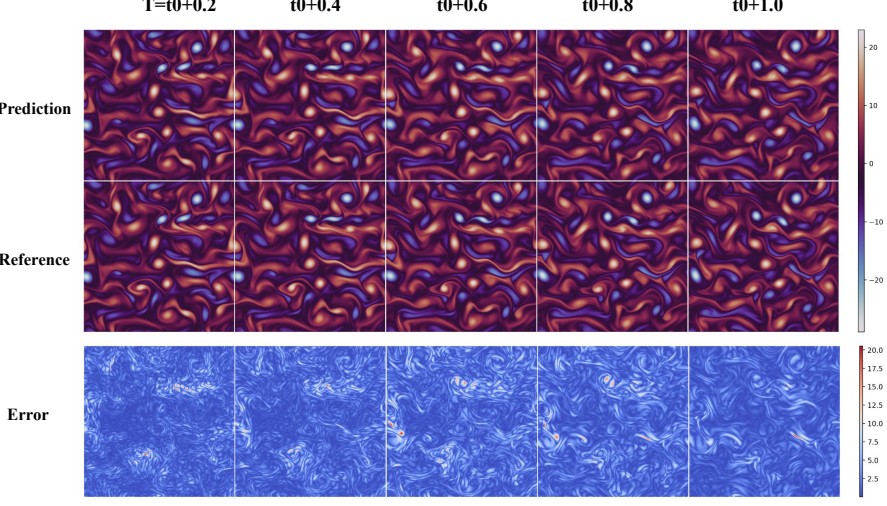

Figure 17: Sample 2 of 2D Kolmogorov flow.

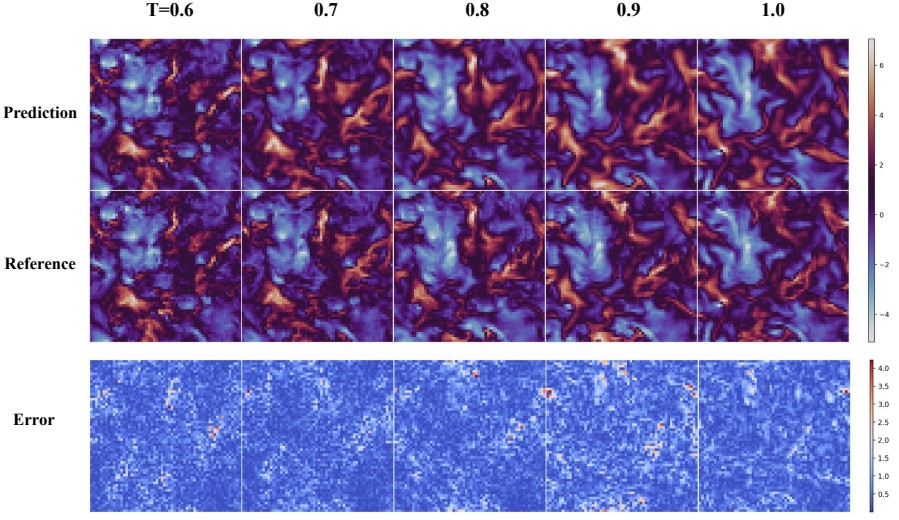

Figure 18: Pressure in 3D isotropic turbulence.

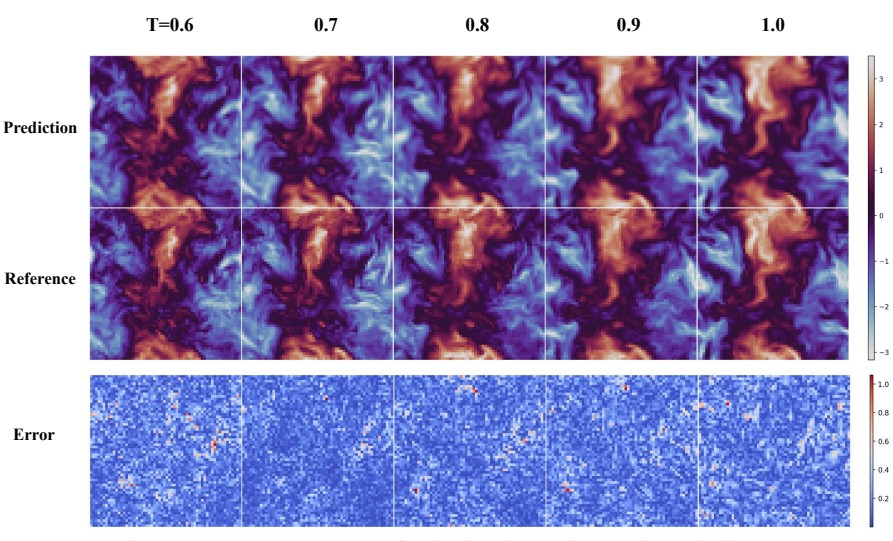

Figure 19: $x$-component of velocity in 3D isotropic turbulence.

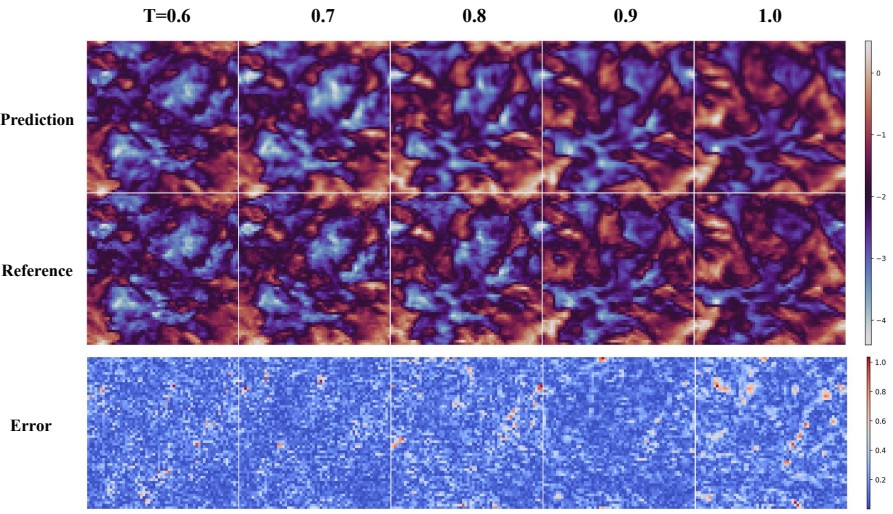

Figure 20: $y$-component of velocity in 3D isotropic turbulence.

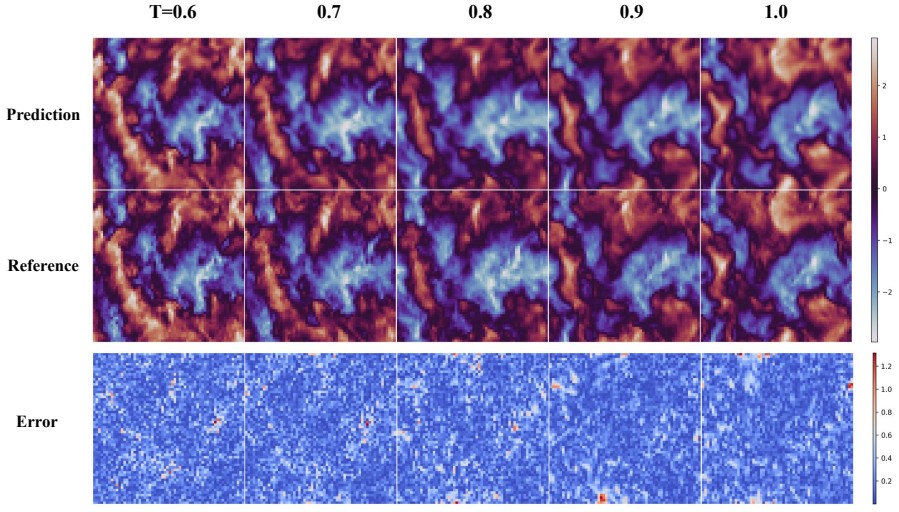

Figure 21: $z$-component of velocity in 3D isotropic turbulence.

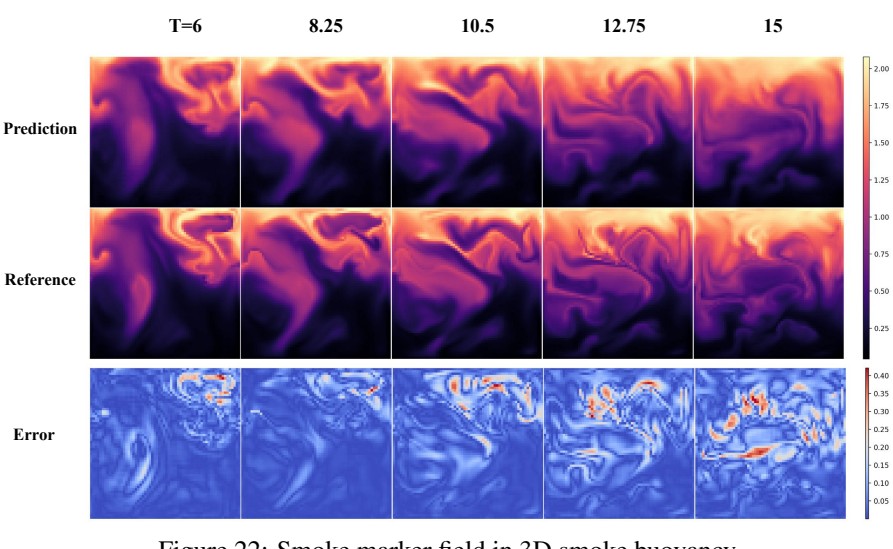

Figure 22: Smoke marker field in 3D smoke buoyancy.

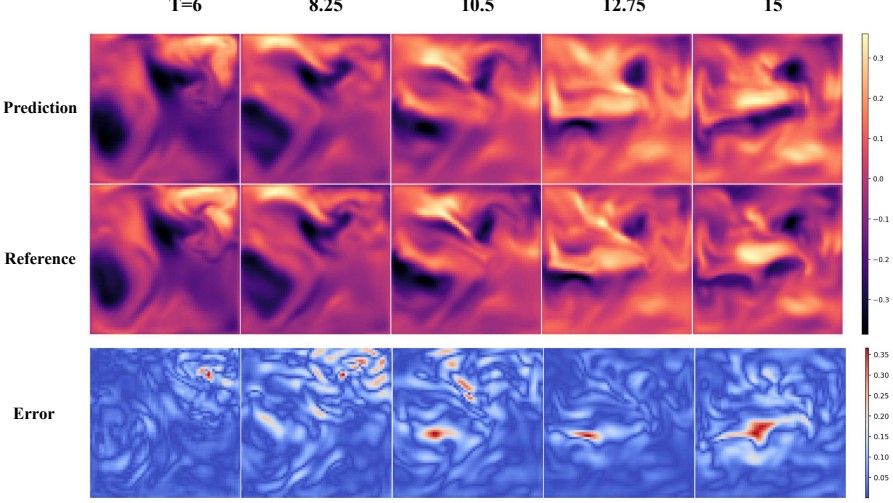

Figure 23: $x$-component of velocity in 3D smoke buoyancy.

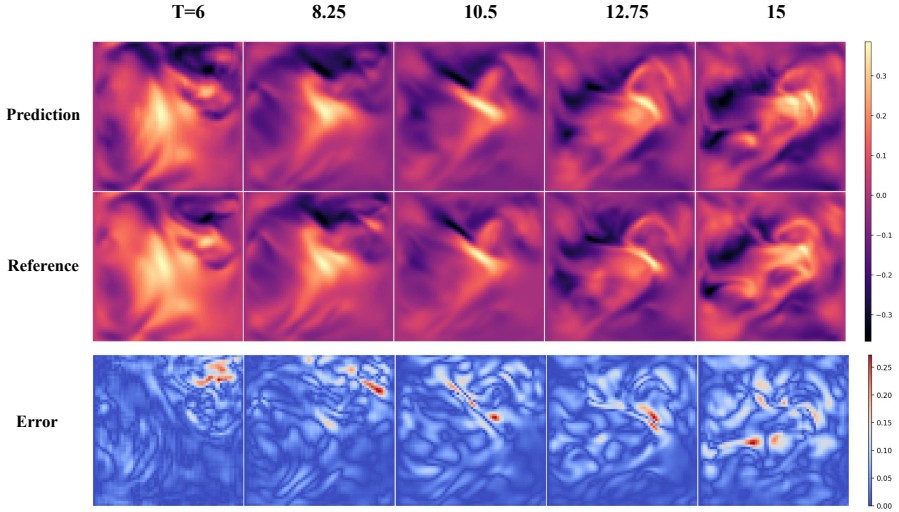

Figure 24: $y$-component of velocity in 3D smoke buoyancy.

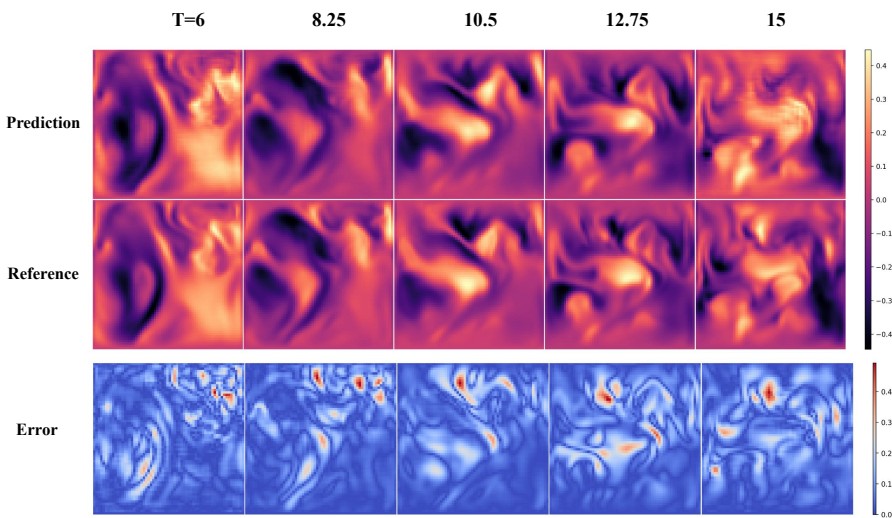

Figure 25: $z$-component of velocity in 3D smoke buoyancy.

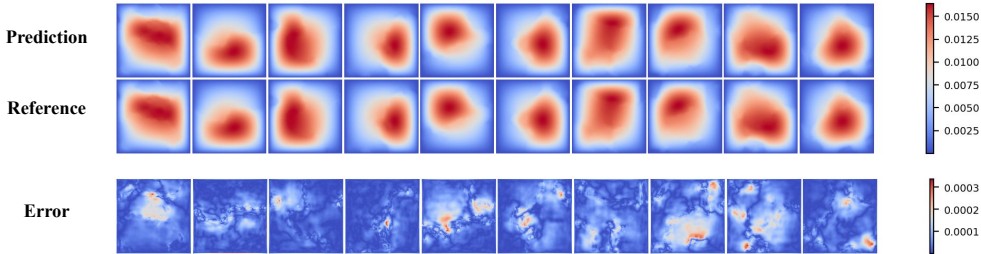

Figure 26: Flow field of 2D Darcy flow.

# G   Broader impact

The numerical simulation of PDEs is of extensive application in various fields, such as manufacturing, weather forecasting, and engineering design. Meanwhile, Transformer has shown promising performance on a wide range of data-driven applications including PDE modeling. Our work can help improve the stability and computational efficiency of the attention-based PDE surrogate models. Our experiments demonstrate that the proposed model serves as an efficient surrogate for numerical solvers, maintaining a balance between accuracy and efficiency, and thus pushing the Pareto front of accuracy-efficiency. However, as there exists a large variety of PDEs and each with very unique properties, there is no guarantee that one type of data-driven model can rule all. Additionally, just like most concurrent works on neural PDE solvers, the long-term stability of the proposed model cannot be guaranteed. Therefore it is important to acknowledge the limitations and potential risks associated with the application of neural PDE solvers.

Apart from enriching the existing architecture design choice of attention-based models, our work also has the potential to be combined with other neural PDE solvers design formulas (e.g., explicitly take into account the relationship between different output variables), or common neural network architectures (e.g., U-Net).

# H   Schematic of Axial Transformer and FactFormer

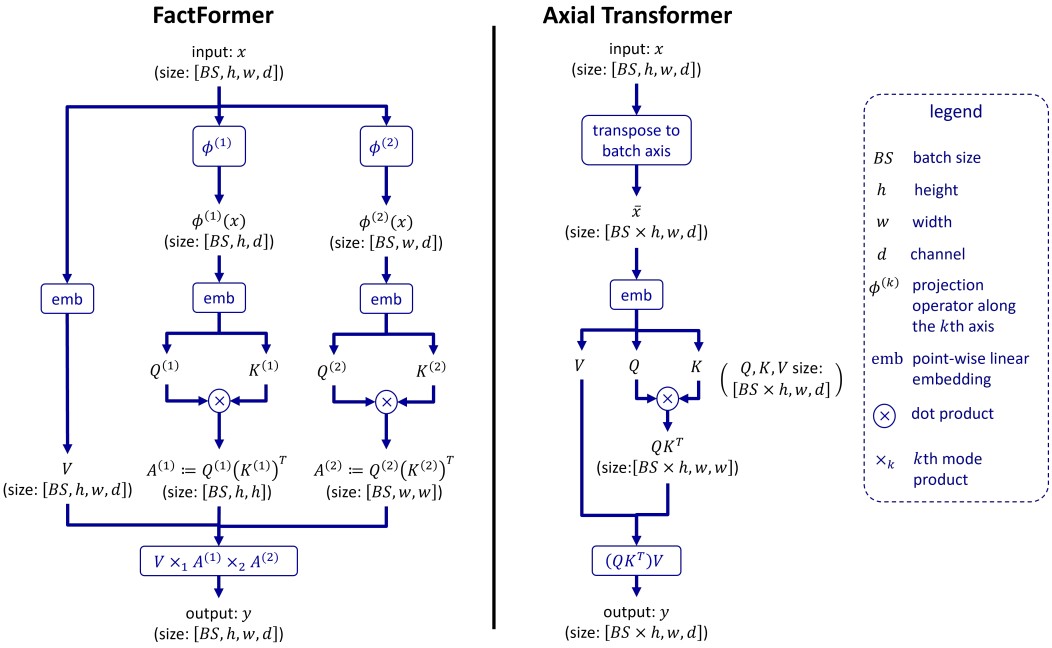

Figure 27: An illustrative example of FactFormer and Axial Transformer applying to 2D input data, with some details such as positional encoding, multi-head mechanism and softmax in Axial Transformer omitted for simplicity. For Axial Transformer, column-wise attention block is shown as an example.

