# OpenReview forum: "Scalable Transformer for PDE Surrogate Modeling"
_NeurIPS.cc/2023/Conference — NeurIPS 2023 poster_

### Official Review · Reviewer_TtUN · 2023-06-15

**Soundness:** 4 excellent
**Presentation:** 4 excellent
**Contribution:** 3 good
**Rating:** 7
**Confidence:** 5

**Summary:**

The paper presents a high-dimensional function decomposition technique for mitigating the complexity encountered by Transformers when dealing with high-dimensional data. The authors have developed a clear logical argument and a comprehensive review of the relevant literature. It conducts extensive experiments on several benchmark 3D problems governed by NS equations. It uses latent marching which shows better empirical performance compared with autoregressive models. However, the paper's novelty appears to be somewhat limited, particularly as similar ideas were presented at ICLR the previous year, including Factorized Fourier Neural Operator (FFNO) and Tensorized FNO.  Overall, the paper lies on the borderline. It presents a clear motivation and technical soundness, and has shown some empirical advantages. However, the limitations highlighted above, particularly the lack of comprehensive experimental comparisons and some confusing notations, hinder its overall quality. I lean towards accepting this paper if the authors could strengthen the experimental results and address the raised concerns.


**Strengths:**

1. The motivation for the research is clear, with the Introduction section being easy to read and understand.
2. From a technical perspective, the paper appears sound, with a well-articulated methodology that would be easily replicable, especially if the authors decided to open-source their code and datasets.
3. Empirically, the proposed method has shown some advantages.

**Weaknesses:**

1. The experimental section is somewhat weak. The authors considered 3D datasets only within regular geometric areas and did not compare their approach with many of the available baselines such as U-FNO, FFNO, HT-Net, Tensorized FNO, and Wavelet NO. The paper's baselines only included standard FNO and ResNet, making it challenging for readers to ascertain whether the Transformer-based approach indeed excels in high-dimensional data processing. I suggest the authors incorporate additional representative baselines (at least one or two) during the rebuttal period.
2. The use of notations in the paper is a little confusing. Does 'd' represent the dimension of the hidden layer? How is it selected? It seems like it would be beneficial to annotate some of the intermediate variables' dimensions or shapes, as the current description in the paper is not very clear.
3. In equation (7), the 'v' kernel appears to still increase exponentially with the dimension, which begs the question, how is this overcome the curse of dimensionality?
4. While the proposed method seems to involve some sort of tensor decomposition on the data, the paper does not investigate the relationship between these processes in detail. It would be beneficial to explore this theoretically, as it could shed light on the model's representational capacity. After all, certain tensor decompositions have theoretical guarantees for approximating a high-dimensional tensor.
5. I suggest you to include the following closely related work in the reference if you have not mentioned them in the paper. FFNO, HT-Net and Tensorized FNO are proposed in the last year to deal with high dimensional data. WNO, U-FNO are also popular architectures for operator learning. And you might need to consider citing a more relevant survey that list many related works on neural operators.

Ref:
1. Factorized Fourier Neural Operators (https://arxiv.org/abs/2111.13802)
2. Multi-Grid Tensorized Fourier Neural Operator for High Resolution PDEs (https://openreview.net/pdf?id=po-oqRst4Xm)
3.  HT-Net: Hierarchical Transformer based Operator Learning Model for Multiscale PDEs(https://openreview.net/forum?id=UY5zS0OsK2e)
4. Multiwavelet-based Operator Learning for Differential Equations (https://arxiv.org/abs/2109.13459)
5. U-FNO -- An enhanced Fourier neural operator-based deep-learning model for multiphase flow (https://arxiv.org/abs/2109.03697)
6. Physics-Informed Machine Learning: A Survey on Problems, Methods and Applications (https://arxiv.org/abs/2211.08064)

**Questions:**

None.

**Limitations:**

None.

---

> ### Author Rebuttal · Authors · 2023-08-09
>
> We thank the reviewer TtUN for the very detailed comments and suggestions. We also appreciate the recognition on our work.
> Here we would like to address your concerns and questions as below.
>
> ---
> > *The experimental section is somewhat weak......I suggest the authors incorporate additional representative baselines (at least one or two) during the rebuttal period.*
>
> We appreciate reviewer's suggestion on making the experiment section more comprehensive. The primary reason for choosing FNO and Dil-ResNet as main baselines is that FNO is a represenative neural-PDE sovler that has shown decent performance on a wide array of problems, while
> DilResNet has shown good accuracy [1, 2] on the class of fluid problems investigated in this work. Nevertheless, following the reviewer's suggestion, we added experiments on Multi-wavelet NO, Factorized-FNO, and
> Tensorized-FNO in our newly uploaded PDF (**Table 2,3,4**). In addition, we also added experiment results (**Table 1**) of our model on datasets from FNO,
> which are used in many other neural-PDE solver literatures. Due to the one page limit we only listed FNO/Linear Transformer/Dil-ResNet for reference, but we will incorpate more results from relevant literatures in the future.
>
> [1] Learned Simulators for Turbulence, ICLR 2023
>
> [2] Towards Multi-spatiotemporal-scale Generalized PDE Modeling, 2023
>
> ---
> > *The use of notations in the paper is a little confusing. Does 'd' represent the dimension of the hidden layer? How is it selected?
> > It seems like it would be beneficial to annotate some of the intermediate variables' dimensions or shapes, as the current description in the paper is not very clear.*
>
> Yes, 'd' can be considered as the hidden dimension in the network. We set it to 128 by default and we invite reviewer to check out **Figure 3** in our **Appendix** where we show that the model's performance
> can be further improved if we adopt a larger dimension (*We apologize for a typo in Figure 3's x-axis, the correct labelling should be: 128 -> 64, 64 -> 128*). We also added an illustrative diagram in the newly uploaed PDF (**Figure 2**) with annotated tensor shapes.
>
>
> ---
> > *In equation (7), the 'v' kernel appears to still increase exponentially with the dimension, which begs the question, how is this overcome the curse of dimensionality?*
>
> It is true that the 'v' has a shape that grows exponentially and we actually acknowledged it in the Conclusion section that our model is still not free from the curse of dimensionality. Our main goal is to alleviate the curse of dimensionality when using attention to parametrize the learnable kernel integral. More specifically, we exploit the low-rank structure of full-attention matrix and replace it with a set of much smaller attention matrices, which improves the computational efficiency and also the numerical stability when applying to higher-dimensional problems with a large number of grid points.
>
> ---
> > *While the proposed method seems to involve some sort of tensor decomposition on the data, the paper does not investigate the relationship between these processes in detail. It would be beneficial to explore this theoretically, as it could shed light on the model's representational capacity. After all, certain tensor decompositions have theoretical guarantees for approximating a high-dimensional tensor.*
>
> We thank reviewer's suggestion to explore the theoretical property of the decomposition process. While SVD-based methods (e.g. Tucker/CP decomposition) come with a strong theoretical foundation, they are not used in our work as it is prohibitively expensive to compute the online SVD for high-dimensional input tensor in each layer during training iterations. Instead, the decomposition from high-dimensional tensor to vector is accomplished through the cheap (yet effective) learnable projector we proposed. Investigating the theoretical capacity of the proposed projection method can be an interesting future direction.
>
> Following this, we would like to also elaborate a bit more about the difference between existing works and our work in terms of how the kernel is parameterized and computed.
>
> * Tensorized-FNO: Apply low-rank decomposition (e.g. Tucker/CP/Tensor-train) to the dense weight in spectral convolution layer and store them in the factorized form.
> * Factorized-FNO: Apply 1D spectral convolution along each dimension separately.
> * FactFormer: Use learnable projection to project tensor into vectors along different axes, and then use these projected vectors to compute data-dependent kernel.
>
> In general, our work explored a new way to compute *data-dependent* kernel in a multi-dimensional factorized way.
>
> ---
> > *I suggest you to include the following closely related work in the reference if you have not mentioned them in the paper...*
>
> We thank the reviewer for the valuable references and we will include them in the future version of the manuscript (and camera-ready version if the paper gets accepted).

---

> ### Comment · Reviewer_TtUN · 2023-08-13
> **Feedback**
>
> Thanks for the response from the authors. Most of my concerns are resolved and I have increased the rating.

---

### Official Review · Reviewer_EAHP · 2023-07-07

**Soundness:** 3 good
**Presentation:** 2 fair
**Contribution:** 3 good
**Rating:** 5
**Confidence:** 5

**Summary:**

This paper uses separable axis attention to reduce the complexity from being exponential in spatial dimension to linear.

**Strengths:**

NA

**Weaknesses:**

The standard Transformer has two components: attention and MLP. This paper only addresses the attention. But the MLP complexity O(Nd^2) would still make the overall complexity exponential in spatial dimension. This seems to defeat the purpose of the paper.

By design, the attention without softmax has rank d, where d is the hidden size and is typically much smaller than the sequence length N. This is shown in Fig 8a. For A^(1) and A^(2) in Fig 8b and 8c, the sequence length along each separate axis is much smaller. This gives the impression that matrices A^(1) and A^(2) are less rank deficient. But this does not mean that the proposed attention is better than vanilla attention.

**Questions:**

Table-4 only reports the encoding or attention layer runtime. What is the overall runtime including MLP layers?

---

> ### Author Rebuttal · Authors · 2023-08-09
>
> We thank reviewer EAHP for the efforts spend on reviewing our work. However, we believe there are some misunderstanding and misinterpretation about our work and we would like to
> provide some clarifications below.
>
> ---
> > *The standard Transformer has two components: attention and MLP. This paper only addresses the attention. But the MLP complexity O(Nd^2) would still make the overall complexity exponential in spatial dimension. This seems to defeat the purpose of the paper.*
>
> We believe the reviewer has misunderstood the goal of the proposed model. The goal of the proposed approach is to alleviate the curse-of-dimensionality of applying attention to higher-dimensional PDE problems and our approach is not completely free from the exponential complexity in spatial dimension (we actually acknowledge that our model requires evaluating function value at all $N$ grid points in the conclusion section). We achieve this goal by exploiting the low-rank structure of the kernel matrix $A=QK^T$, replacing it with the product of a set of much smaller axial kernels. Through extensive experiments, we have showcased that our proposed approach enjoys notably better computing efficiency and numerical accuracy than the model using softmax-free attention.
>
> In addition, despite the asymptotic bound of MLP being $O(Nd^2)$, it is not the major bottleneck of attention-based PDE solvers in practice. For example, on the $128 \times 128$ problem, MLPs only account for roughly 16\% of the total calculation time for FactFormer, and less than 5\% for Linear Transformer.
>
> ---
> > *Table-4 only reports the encoding or attention layer runtime. What is the overall runtime including MLP layers?*
>
> We actually include most MLPs' runtime in Table-4 and we will make this point clearer in the future version of the manuscript. The only MLP that has been excluded is the one that's used for propagating dynamics in the latent space. More specifically, the model has 4 attention layers followed by 4 MLPs, and an additional propagating MLP. The *Enc. time* corresponds to the total runtime of  4 attention layers + 4 MLPs. The *Prop. time* corresponds to calling the propagating MLP 4 times to propagate system state from $z_{t}$ to $z_{t+4}$.
>
>
> ---
> > *By design, the attention without softmax has rank d, where d is the hidden size and is typically much smaller than the sequence length N. This is shown in Fig 8a. For A^(1) and A^(2) in Fig 8b and 8c, the sequence length along each separate axis is much smaller. This gives the impression that matrices A^(1) and A^(2) are less rank deficient. But this does not mean that the proposed attention is better than vanilla attention.*
>
> The experiment here is to provide heuristic motivation for replacing the full-attention with axial factorized attention. When parametrizing the learnable kernel integral with vanilla attention, the kernel
> matrix $A=QK^T$ has very low rank by design (since $rank(AB)\leq \min[rank(A), rank(B)]$ and as reviewer points out, $rank(Q), rank(K)$ is upper bounded by channel number $d$). This motivates us to propose a axial-factorized kernel integration scheme and replace $A$ with a set of much smaller (but higher-rank) matrices $A^1, A^2, ..., A^n$. And the experiment results confirm that $A^1, A^2, ..., A^n$ indeed exhibit higher-rank structures after training.
>
> Based on reviewer's feedback, we will revise the description here to avoid ambiguous impression. Essentially, the proposed approach doesn't differ from vanilla linear attention in terms of how to calculate attention score (they are both dot product attention), the difference is how to parameterize and compute the kernel in the learnable kernel integral transform with attention.

---

> ### Comment · Reviewer_EAHP · 2023-08-20
>
> Thanks the authors for the clarification and answering my questions. I have increased my ratings.

---

### Official Review · Reviewer_y1WA · 2023-07-07

**Soundness:** 4 excellent
**Presentation:** 2 fair
**Contribution:** 3 good
**Rating:** 7
**Confidence:** 5

**Summary:**

This paper proposes a new method to scale Transformer-based models for PDE surrogate modeling to higher dimensions. The new method combines several "tricks" from previous works, and most importantly, uses factorization to further reduce the computational cost of training and evaluating Transformer-based operator learners, while not sacrificing the accuracy too much, and even achieves superior performance in certain benchmark problems. Even though the idea of using factorization (tensor decomposition) for end-to-end problems that feature a spectrum decay and/or sparse structures is not new, I check the Transformer-based PDE modeling papers and this is the first time someone implemented it.

I think this is a worthy contribution to the literature, however, there are apparent weaknesses in the paper as well (detailed below). I lean toward acceptance if the author can address or answer some of the concerns below.

**Strengths:**

- The authors proposed a simple way to reduce the computational cost of linearization of attention further. The method is actually quite simple by using a different projection and iteratively evaluating the attention "integral", which is essentially tensor decomposition when the un-integrated kernel matrices are moved to the innermost integral written together. As the reason behind tensor decomposition makes more mathematical sense than the one featured in Axial Transformer.
- Dramatic savings versus even the linear attention.
- I actually appreciated that the authors spilled some techniques that are usually deemed "tricks" explicitly in Section 3.3, and used them across all baselines,  instead of hiding them in the source code as some works on Transformers.
- The authors enhanced many existing PDE benchmarks (in the anonymized GitHub repo), and created several new ones that test the scalability of the models.

**Weaknesses:**

- The biggest weakness is perhaps some lack of theoretical foundation, but I guess this is fine for a methodology paper.
- The saving of computation in the presentation of formula (7) may not be that obvious to the community of Transformer research, who may not be that familiar with the integral representation. Especially considering the presentation of (7) uses $n$-D, but compares that in (2), which is for a 1-D problem.
- The presentation in laying out the factorization could use some improvement for people more familiar with linear algebra than differential equations. For example, on line 179,
- If only the author could add a more informative diagram comparing the difference between the factorized approach vs Axial Transformer.

### Misc small typos
- line 144: there should be space between RoPE and its reference.
- line 152: this is just a suggestion, the tilde notation could be clearer if it is referred back to equation (3), saying something like "$\tilde{\mathbf{z}}$ denotes a matrix $\mathbf{z}$ whose row vectors $\mathbf{z}_j$ are RoPE encoded as in (3)".
- line 154: "learnable projection" -> "learnable projections".
- line 209: in (9), $Z$ should be set to $\operatorname{Att}(U)$.

**Questions:**

- In the comparison of SVD for attention matrices, what is the value of $k$? What specifically are $A^{(1)}$ and $A^{(2)}$? From different model problems?
- In Section 4.2, the FNO models have significantly higher parameter counts. Is there a specific reason for this?


**Limitations:**

The author properly acknowledged the curse of dimensionality. Moreover, the dimension-bound factorization limits the domain type to tensor-product type (i.e., rectangular).

---

> ### Author Rebuttal · Authors · 2023-08-09
>
> We thank reviewer y1WA for the helpful comments and greatly appreciate your recognition on our work. Here we would like to address your concerns and questions as below.
>
>
> ---
> > *The saving of computation in the presentation of formula (7) may not be that obvious to the community of Transformer research, who may not be that familiar with the integral representation. Especially considering the presentation of (7) uses n-D,
> >  but compares that in (2), which is for a 1-D problem.*
> >
> > *The presentation in laying out the factorization could use some improvement for people more familiar with linear algebra than differential equations...*
>
> We appreciate reviewer's suggestion on making this paper more accessible to the broader audience. In (2), we are trying to define a general domain $\Omega$ without specifying its dimension, it can also be greater than $1$-D. In the next version of the manuscript, we will make this point clearer, and incorporate a more concrete expression explaining the difference between the standard attention and axial factorized ones, that is our proposed scheme amounts to replacing $z=AV$ with tensor-matrix product $z=V\times_1 A^1 ... \times_n A^n$ where $A^1, ..., A^n$ are much smaller matrices than full attention matrix $A$.
>
>
> ---
> > *If only the author could add a more informative diagram comparing the difference between the factorized approach vs Axial Transformer...*
>
> In the newly uploaded PDF (**Figure 2**), we added a diagram that describes the differences between how the proposed approach and Axial Transformer deal with 2D input.
> We will also add this to the Appendix of the paper in the future version.
>
> ---
> > *Misc small typos...*
>
> Thanks for the catch, we will correct them in the next version of the manuscript.
>
> ---
> > *In the comparison of SVD for attention matrices, what is the value of k? What specifically are A^{(1)} and A^{(2)}?*
>
> $k$ is the index of the singular value (used to compute the faction of singular values in the figure, i.e. k divided by total number of singular values).
> $A^{(1)}, A^{(2)}$ are sub-attention matrices corresponding to two axes $y$ and $x$, respectively.
>
> ---
> > *In Section 4.2, the FNO models have significantly higher parameter counts. Is there a specific reason for this?*
>
> This is because the number of parameters of FNO grows exponentially with respect to the problem dimension. The parameter count of the kernel in each FNO kernel integral layer is $O(M^nd^2)$, where $M$ is the number of truncated modes, $n$ is the number of the problem dimension. For example, using 16 modes and a hidden dimension of 128 in a FNO3D layer will result in parameter count of $O(16^3 \times 128^2)$.

---

> > ### Comment · Reviewer_y1WA · 2023-08-14
> >
> > The reviewer appreciates the author adding many experiments and the diagram in such a short response window. I think it is a worthy addition to the literature, for both operator learning and Transformer architecture research community. Therefore, I raised the score from 6 to 7.

---

### Official Review · Reviewer_UCqx · 2023-07-24

**Soundness:** 3 good
**Presentation:** 3 good
**Contribution:** 3 good
**Rating:** 6
**Confidence:** 5

**Summary:**

The proposed work presents a scalable transformer architecture for modeling partial differential equations (PDEs). The input function is linearly projected to multiple functions with one-dimensional domains using a learnable integral projection operator. The attention mechanism is then applied to these projected functions. This factorized attention mechanism requires less time and memory. The proposed FactFormer shows comparable or better performance while requiring fewer parameters and less memory.

**Strengths:**

1. The proposed method is effective and scalable, making it suitable for high-resolution PDE modeling.
2. The paper is well-written, and the method is described in detail.

**Weaknesses:**

1. The experiments in this work use fluids with very high viscosity, resulting in simulations of flows with low Reynolds numbers. As the model uses linear projection operations and performs softmax-free attention in the projected functions, it is crucial to demonstrate the method's effectiveness in modeling complex systems, such as flows with high Reynolds numbers. For instance, the set of experiments performed in FNO to model the Navier-Stokes equation with varying viscosities.

**Questions:**

1. is there any non-linearity or activation function used in the architecture?
2. How well does the model perform in the zero-shot super-resolution task compared to the FNO baseline?

---

> ### Author Rebuttal · Authors · 2023-08-09
>
> We thank the reviewer UCqx for the helpful comments and suggestions on improving the paper. Here we would like to address your concerns and questions as below.
>
> ---
> > *The experiments in this work use fluids with very high viscosity, resulting in simulations of flows with low Reynolds numbers. As the model uses linear projection operations and performs softmax-free attention in the projected functions, it is crucial to demonstrate the method's effectiveness in modeling complex systems,
> > such as flows with high Reynolds numbers. For instance, the set of experiments performed in FNO to model the Navier-Stokes equation with varying viscosities.*
>
> Based on reviewer's suggestion, we have added experiments on datasets from FNO with varying viscosities (please refer to **Table 1** in the newly uploaded PDF). We observe that FactFormer also has competitive performance on these datasets. In addition, the 2D Kolmogorov dataset in our work also has a relatively low viscosity (with Re=1000).
>
> ---
> > is there any non-linearity or activation function used in the architecture?
>
> Yes, as shown in Figure 2 of the main manuscript, we apply a pointwise feedforward network to the output of each attention layer, which uses GELU activation function.
>
> ---
> > How well does the model perform in the zero-shot super-resolution task compared to the FNO baseline?
>
> The proposed model can also generalize to unseen resolutions like FNO. Below we provide an experiment on Darcy flow:
>
> (Metric is relative L2 error)
> |Resolution|211(train)|421|
> |----------|----------|---|
> |FNO2D|0.0073|0.0141|
> |FactFormer|0.0058|0.0133|

---

> > ### Comment · Reviewer_UCqx · 2023-08-16
> > **Response to the Rebuttal**
> >
> > I thank the authors for their response. The proposed method also achieves comparable performance in modeling fluid flow with high Re. I am increasing the score.

---

> > > ### Comment · Reviewer_UCqx · 2023-08-19
> > > **Further clarification**
> > >
> > > I have one additional question regarding the experiments.
> > >
> > > **Question**:
> > > 1.  How many data points are used for training and testing in the each of the experiments?

---

> > > > ### Author Response · Authors · 2023-08-20
> > > > **Reply to Reviewer UCqx**
> > > >
> > > > Thanks for the additional comments. Below are the number of train/test samples we used through experiments.
> > > >
> > > > > Newly added experiments (Table 1 in the new pdf):
> > > >
> > > > We follow the setting from the original FNO paper. For different cases in Navier-Stokes equation, we use 1000 samples (trajectories) for training and 200 for testing. For Darcy flow, we use 1000 samples for training and 100 for testing.
> > > >
> > > > > Experiments in the main manuscript:
> > > >
> > > > * 2D Kolmogorov flow: We use 100 trajectories for training and 20 trajectories for testing. Each trajectory contains 160 frames with $\Delta t=0.0625$.
> > > > * 3D Isotropic turbulence: We use 1000 trajectories for training and 100 trajectories for testing. Each trajectory contains 20 frames with $\Delta t= 0.05$.
> > > > * 3D Smoke buoyancy: We use 2000 trajectories for training and 200 trajectories for testing. Each trajectory contains 20 frames with $\Delta t=0.75$.
> > > >
> > > > We will also include the number of samples in the next version of the manuscript.

---

### Official Review · Reviewer_hvcj · 2023-07-25

**Soundness:** 2 fair
**Presentation:** 3 good
**Contribution:** 2 fair
**Rating:** 5
**Confidence:** 3

**Summary:**

This paper proposes Factorized Transformers (FactFormer) for surrogate modeling of partial differential equations (PDEs). FactFormer first projects the input function to multiple one-dimensional projected functions, which are then used for multi-dimensional factorized attention. The factorized attention greatly reduces the model complexity, while still achieving competitive performance on three turbulence problems.

**Strengths:**

1. Improving the efficiency of transformer is important to the development of transformers in PDE modeling.
2. The proposed FactFormer achieves superior model efficiency with competitive performance.

**Weaknesses:**

1. The design of learnable projection and factorized kernel integral may limit the model capacity, especially for problems that do not have such underlying low rank structure. For example, the projection block basically gets $n$ projections of the input function on $n$ dimensions, which could drop the crucial information about the input function.
2. In principle, full attention should have better model capacity than factorized attention. However, the experimental results in this paper show that full attention is much worse. The authors hypothesize that it is due to the instability of rollout. An error trend plot of full attention and factorized attention is need to verify this hypothesis.
3. The authors hypothesize that CNN variants are better than FNO on turbulence problems because CNN filters capture high-frequency patterns but FNO truncates the high-frequency modes.  However, the nonlinear activation function and the local linear transform branch in FNO allows FNO to model high-frequency patterns.  [1] has shown it can efficiently approximate operators in incompressible Navier-Stokes equations. Further justification and ablation study is needed to support this hypothesis.
4. The inference speed of FactFormer is 1.5-1.8x slower than FNO while the number of parameters is much smaller. Why is that? Can you also report the FLOPs of the FactFormer and compare it with other baseline models?
5. Overall, the accuracy of FactFormer is not consistently better than Dil-ResNet.

Minors:
1. Line 179: $L_N$ should be $L_n$.
2. Line 302: "time cost for ..." -> "time cost of ..."
3. Line 315: "time cost by ..." -> "time cost of ..."

[1]: Kovachki, Nikola, Samuel Lanthaler, and Siddhartha Mishra. "On universal approximation and error bounds for Fourier neural operators." The Journal of Machine Learning Research 22.1 (2021): 13237-13312.

**Questions:**

1. How is the error trend plot computed exactly? I thought it is accumulative error but the error is not monotonically increasing.
2. Can you also compare the model efficiency with tensorized FNO, which leverages the tensor factorization to improve efficiency of the original FNO?
3. Section 4.3 claims that Factorized attention has higher ranks so it is more efficient. Can you elaborate more on how you jump to that conclusion?

**Limitations:**

Listed in weaknesses and questions sections.

---

> ### Author Rebuttal · Authors · 2023-08-09
>
> We thank the reviewer hvcj for the insightful comments and suggestions on the paper. Here we would like to address your concerns and questions as below.
>
> ---
> >*The design of learnable projection and factorized kernel integral may limit the model capacity, especially for problems that do not have such underlying low rank structure.
> > For example, the projection block basically gets n projections of the input function on n dimensions, which could drop the crucial information about the input function.*
>
> The projection part exploits the low-rank structure and drops information of other axis, but the value branch in the attention layer preserves the information of all dimensions and then a pointwise feed-forward network is applied to the output of the attention layer (somewhat similar to the local branch in FNO).
>
> ---
> > *The authors hypothesize that it is due to the instability of rollout. An error trend plot of full attention and factorized attention is needed to verify this hypothesis.*
>
> We've added a figure of the average rollout error trend of Linear/Factorized attention in the newly uploaded PDF (**Figure 1**).
>
> ---
> > *The authors hypothesize that CNN variants are better than FNO on turbulence problems because CNN filters capture high-frequency patterns but FNO truncates the high-frequency modes. However, the nonlinear activation function and the local linear transform branch in FNO allow FNO to model high-frequency patterns [1]...*
>
> We appreciate the reviewer for the valuable reference. We will revise our description of the high-frequency bias hypothesis and incorporate the reference in the next version of the paper. We agree that theoretically FNO has the capacity to capture high-frequency patterns, but in practice, this also depends on the model's training dynamics and the problems it is applied to. [1] poses that learning the linear transform of Fourier coefficients can potentially suffer from spectral bias [2] and choose to preserve lower frequency modes during training.
>
> [1] Incremental Fourier Neural Operator, 2022.
>
> [2] On the spectral bias of neural networks, 2019.
>
> ---
> > *The inference speed of FactFormer is 1.5-1.8x slower than FNO while the number of parameters is much smaller. Why is that? Can you also report the FLOPs of the FactFormer and compare it with other baseline models?*
>
> The major reason is that the number of parameters of FNO grows exponentially with respect to the problem dimension while FactFormer does not (the projection operator of FactFormer grows linearly). The parameter count of the kernel in each FNO kernel integral layer is $O(M^nd^2)$, where $M$ is the number of truncated modes, $n$ is the number of the problem dimension. For example, using 16 modes and a hidden dimension of 128 in a FNO3D layer will result in parameter count of $O(16^3 \times 128^2)$. We also list the FLOPs (on 3D grid) measured by DeepSpeed library below:
>
> |Model|Tensorized-FNO|Factorized-FNO|FNO|Dil-ResNet|Linear Transformer|FactFormer|
> |-----|--------------|--------------|---|----------|-----------|-------------------|
> |GFLOPs |33 | 101 | 33 |3501|1685|596|
>
> ---
> > *How is the error trend plot computed exactly? I thought it is accumulative error but the error is not monotonically increasing.*
>
> Frame-wise error is shown in the error trend plot. LM models predict multiple future steps within one model call and thus result in the error not always monotonically increasing within a prediction window.
>
> ---
> > *Can you also compare the model efficiency with tensorized FNO...*
>
> In the newly uploaded one-page PDF, we added tensorized FNO to the runtime comparison (**Table 5**) and we also tested it on 2D problem (**Table 2**).
>
>
> ---
> > *Section 4.3 claims that Factorized attention has higher ranks so it is more efficient. Can you elaborate more on how you jump to that conclusion?*
>
> The low-rank property of large-sized full attention matrix $A$ hints that it is possible to simplify the original kernel integral computation without too much loss of information.
> We approach this simplification by replacing the original kernel integral with an axial factorized integral which only involves a group of much smaller attention matrices $A^1, ..., A^n$ (with higher rank), which improves the computational efficiency. Based on the reviewer's feedback, we will revise the discussion in section 4.3 to improve the clarity.
>
> ---
> > *Spotted typos...*
>
> We thank the reviewer for the catch. We will correct them in the next version of the manuscript.

---

### Official Review · Reviewer_YpQV · 2023-07-27

**Soundness:** 2 fair
**Presentation:** 3 good
**Contribution:** 3 good
**Rating:** 4
**Confidence:** 2

**Summary:**

The authors propose FactFormer, a neural PDE solver that is based upon the transformer, but scales much better with the dimensionality of the PDE problem.   The authors compare FactFormer to several other neural PDE solvers on three different problems and find that Factformer achieves similar performance to state-of-the-art PDE solvers, but while offering lower computational costs.  This property may also allow FactFormer to scale to high-dimensional problems than existing PDE solvers.

**Strengths:**

1) The paper is generally well-written
2) The authors use three challenging PDE problems, and they compare their approach to two state-of-the-art existing models, providing reasonably good empirical evidence for the performance of their approach.
3) The authors do provide some analysis comparing FactFormer to other Attention mechanisms, which is helpful for proving their advantage.

**Weaknesses:**

1) (Major) Upon inspection, it doesn't appear that these baseline models (FNO and Dil-ResNet) have been previously applied by other authors to any of the three benchmark problems (with the particular settings used in this work), so that the authors here had to implement and apply the baseline methods themselves.  This introduces a major potential source of bias if the authors, because it is unclear whether the authors made equal effort to optimize the performance (e.g., by adjusting model size, batch sizes, learning rates, etc) of the baseline methods and their proposed approach.  It would be better if the authors could test FactFormer on one or two PDEs where the baseline methods (FNO and Dil-ResNet) have previously been optimized and tested by other authors, helping to ensure a more fair comparison.    For example, why don't the authors use the data/benchmarks from [84], for example?

2) (Major) The main claim of this paper is that attention is good, but we cannot use it because of the computation time.   Therefore we make it more efficient, resulting in FactFormer, but the FactFormer model doesn't perform any better than existing models (e.g,. Dil-ResNet).  Therefore, if there was an advantage to using Transformers for PDE solvers, apparently that advantage disappears when using the factorization in the Factformer.   So then the main contribution here is presumably the computational time of Factformer?  Factformer does seem to be significantly more computationally efficient than Dil-ResNet for Inference - which is great - but what about training time?   If it is not also superior during training then this is a major limitation.

3) (Minor) The main stated contribution of this paper is to extend transformers to solve higher-dimensional problems, rather than outperform other PDE solvers.  Indeed, the Dil-ResNet often outperforms the FactFormer, and therefore there seems to be little, if any, performance advantage against recent PDE solvers.  The authors show some theoretical big-"Oh" analysis of runtime, but these are asymptotic bounds.  In Table 4 the authors report some computation times, but it is just for one problem, and it is unclear what problem they are testing on (perhaps I missed it?).    Since this is a main claimed contribution of this work, it seems quite important to me to report the empirical computation time (e.g., per epoch or iteration) for PDE problems with some varying mesh sizes and dimensionalities, so that we can see how the runtimes compare for some common settings of these parameters.





**Questions:**

1) Per Weakness #2 above, What is the computational cost of training FactFormer compared to Dil-ResNet and FNO?

2) Is there a reason the authors did not use existing benchmark PDE problems to test FactFormer (per my Weakness #1 comment above)?   It is important to me that there is a good reason, or alternatively, that the authors can either (i) provide evidence that they fairly optimized all competing models (this is very hard to do), or (ii) add another benchmark that was used in a prior study, and compare Factformer's performance to that reported for the baselines on that prior study.

If these questions are (convincingly) addressed I am willing to significantly increase my score, potentially to "Accept".

---

> ### Author Rebuttal · Authors · 2023-08-09
>
> We thank the reviewer YpQV for the detailed comments and suggestions on improving the paper. Here we would like to address your concerns and questions as below.
>
> ---
> > *The motivation for creating new datasets.*
>
> One of the major goals of our work is to improve the scalability of attention-based PDE learning model on higher-dimensional PDE problems and many existing benchmarks consider only 1D or 2D problems.
> To this end, we created a dataset with two challenging 3D problems and a high-resolution 2D problem to stress-test our newly proposed model and existing neural PDE solvers.
>
> ---
> > *Why don't the authors use the data/benchmarks from [84]?*
>
> To our knowledge, the data and code of [84] are not publicly available yet.
>
> ---
> > *Is there a reason the authors did not use existing benchmark PDE problems to test FactFormer (per my Weakness #1 comment above)? It is important to me that there is a good reason, or alternatively, that the authors can either
> >  (i) provide evidence that they fairly optimized all competing models (this is very hard to do), or (ii) add another benchmark that was used in a prior study,
> >  and compare Factformer's performance to that reported for the baselines on that prior study.*
>
> We appreciate reviewer's suggestion of a fair and reliable benchmark. To address your concern, we have added new experiments of FactFormer
> on commonly used benchmark datasets including 2D Navier-Stokes and 2D Darcy flow from FNO (*please refer to **Table 1** in the newly uploaded one-page PDF*). We observe FactFormer also has competitive performance on these datasets.
> In addition, we plan to open-source the code and datasets proposed in this work in the future to facilitate the development of other models in this area.
>
> ---
> > *Indeed, the Dil-ResNet often outperforms the FactFormer, and therefore there seems to be little, if any, performance advantage against recent PDE solvers.*
>
> We agree with reviewer that our proposed model does not consistently outperform Dil-ResNet. However, we do want to point out that as there exists a large variety of PDEs, there is no guarantee that one type of model can rule them all.
> Prior studies have reported that Dil-ResNet excels on 3D turbulence [1] and 2D smoke buoyancy [2]. In our newly added experiments, we show that our proposed model actually has better accuracy over Dil-ResNet on some other 2D problems.
> In addition, unlike FNO and FactFormer, without changing the architecture, Dil-ResNet's performance deteriorates significantly when the resolution increases (**Table 1 - Darcy flow 2D** in the newly uploaded PDF). In this case, our proposed model has roughly the same level of accuracy under varying resolutions.
>
> [1] Learned Simulators for Turbulence, ICLR 2023
>
> [2] Towards Multi-spatiotemporal-scale Generalized PDE Modeling, 2023
>
> ---
> > *In Table 4 the authors report some computation times, but it is just for one problem, and it is unclear what problem they are testing on (perhaps I missed it?)*
>
> The problem tested is $128 \times 128$ Kolmogorov flow (*mentioned in line 331 of manuscript, we will make this clearer based on reviewer's feedback*). On top of that, we would like to invite reviewer to also check out **Figure 1** and **Figure 2** in the **Supplementary material** , which includes runtime reports under different model sizes and mesh sizes/dimensionalities.
>
> ---
> > *What is the computational cost of training FactFormer compared to Dil-ResNet and FNO?*
>
> We have added the full runtime (forward+backward) in **Table 5** of the newly uploaded pdf. The relative order is similar to the inference time we reported in the manuscript.

---

> > ### Comment · Reviewer_YpQV · 2023-08-14
> > **Thank you for your detailed response, and quick followup questions**
> >
> > I thank the authors for their detailed response to my comments.  I find most of the argumentation and new results convincing.  I have increased my rating by one point   I will consider increasing it by another point or two, but I have one more follow up question regarding the fair comparison of FactFormer with prior competing approaches.
> >
> > In Table 1, the authors added additional experimental results for the Navier-Stokes and Darcy-Flow problems, which I appreciate, and which I believe invariably strengthens the results of the paper.  Per my previous commentary however, these results would be even more convincing (much more in my opinion) if the results of the competing methods (e.g., FNO, Dil-Resnet) on this problem were reported from prior work, where the authors of the competing methods had optimized these models for the problems under study.   With this in mind I have three (hopefully quick ) follow up questions:
> > 1) Is that still the case that you (the authors) implemented the competing models yourself on these new problems (Navier-Stokes and Darcy)?
> > 2) If yes, then why?  e.g., is the data for these problems from prior studies not publicly available so you could run FactFormer on the same data?
> > 3) Even if data from prior studies is not available, how do your results with the competing models compare with those obtained from prior studies?   Are they similar?  If the authors could provide a short description and/or point me to a particular study, and the relevant Table/Figure numbers, that would be helpful.

---

> > > ### Author Response · Authors · 2023-08-14
> > > **Reply to reviewer YpQV**
> > >
> > > Thanks for your consideration and additional comments.
> > >
> > > The results from FNO and Linear Transformer are based on their original paper, except that we re-run FNO's results on Darcy using its improved latest official implementation for a fairer comparison. FNO reported around 1.09e-2 relative error on Darcy in its original paper, whereas we have reported around 0.70e-2 relative error using an updated and improved implementation from the official repo.
> > >
> > > For the official results from FNO and Linear Transformer (LT), we kindly refer reviewer to the following papers:
> > >
> > > * Table 1 (Navier-Stokes) and Table 4 (Darcy flow) in paper: "*Fourier Neural Operator for Parametric Partial Differential Equations*" (arxiv version)
> > >
> > > * Table 2(b) (Darcy flow) in paper: "*Choose a Transformer: Fourier or Galerkin*" (arxiv version)
> > >
> > > * Table 1 (Navier-Stokes) in paper: "*Transformer for Partial Differential Equations’ Operator Learning*" (arxiv version)
> > >
> > > On top of that, we implement and run the experiments for Dil-ResNet as there is no other work reporting its performance on these datasets to our knowledge. We have also uploaded the scripts for reproducing FactFormer on these datasets in the anonymous repo (link in the Section 1 of Appendix).

---

### Author Rebuttal · Authors · 2023-08-09

We would like to thank all the reviewers for their efforts spent reviewing our work and the suggestions on helping us improve our work.

Below we summarize the newly added results that are included in the one-page pdf.

#### **Table 1**:
We tested our proposed model and Dilated-ResNet on two benchmark problems from FNO [1], which are 2D Navier-Stokes equation with different viscosities and 2D Darcy flow with different resolutions.
We also included the result of Linear Transformer [2] for comparison. In general, we observe that our proposed FactFormer also has competitive performance on these problems.
#### **Table 2/3/4**:
We added three new baselines on problems studied in the paper. The newly added baselines are Multi-wavelet neural operator(MWT) [3], and two newly proposed variants of FNO:
Tensorized-FNO [4], Factorized-FNO [5]. Tensorized-FNO(T-FNO) factorizes the weight in a standard FNO layer with tensor decomposition technique. Factorized-FNO (F-FNO) introduces a group of tricks for improving standard FNO and
applying spectral convolution in an axial-factorized way.

*Note: We opt for not including the results of MWT and T-FNO on 3D problems to avoid misleading conclusions.
 The current MWT implementation does not include architecture for applying 3D wavelet transform/bases projection and can only operate on resolutions that are a power of 2.
 The original T-FNO in [4] is applied to 1D/2D steady-state prediction problems. Our direct application of T-FNO on 3D time-dependent problems has a notable degradation compared to FNO.*
#### **Table 5**:
Detailed runtime comparison between different baseline models.
#### **Figure 1**:
Averaged rollout error trend of model using  linear/factorized attention scheme.
#### **Figure 2**:
A diagram comparing Axial Transformer[6] and the proposed model.

We have also addressed the detailed comments of every reviewer in separate replies. Please don’t hesitate to let us know if there is any further question or additional comment.

*References*:

[1] Fourier neural operator for parametric partial differential equations, 2021.

[2] Choose a transformer: Fourier or galerkin, 2021.

[3] Multiwavelet-based operator learning for differential equations, 2021.

[4] Multi-Grid Tensorized Fourier Neural Operator for High Resolution PDEs, 2023.

[5] Factorized Fourier Neural Operators, 2022.

[6] Axial Attention in Multidimensional Transformers, 2019.

---

### Decision · Program_Chairs · 2023-09-21

**Decision:**

Accept (poster)

**Comment:**

After rebuttal and discussions, the reviewers overall lean towards acceptance as the authors have addressed most of the concerns. Given the importance of these discussions and additional experiments in the acceptance of the paper, please carefully finalize them and incorporate the new results, discussions and citations in the paper. The main concern that remains is over the experimental setting. Please carefully report the codebase that was use and the hyper-parameters in all cases.